# Mild anemia and 11- to 15-year mortality risk in young-old and old-old: Results from two population-based cohort studies

**Alessia A. Galbussera[1], Sara Mandelli[1], Stefano Rosso[2], Roberto Zanetti[2], Marianna Rossi[3,4], Adriano Giacomin[5†], Paolo Detoma[6], Emma Riva[1], Mauro Tettamanti[1], Matteo G. Della Porta[3,4], Ugo Lucca[1]\***

1 Laboratory of Geriatric Neuropsychiatry, Department of Neuroscience, Istituto di Ricerche Farmacologiche Mario Negri IRCCS, Milano, Italy, 2 Piedmont Cancer Registry, Centre for Epidemiology and Prevention in Oncology in Piedmont, Turin, Italy, 3 Istituto di Ricovero e Cura a Carattere Scientifico (IRCCS) Humanitas Research Hospital, Milan, Italy, 4 Department of Biomedical Sciences, Humanitas University, Milan, Italy, 5 Epidemiology Unit, Azienda Sanitaria Locale, Biella, Italy, 6 Laboratory of Analysis, Ospedale degli Infermi, Biella, Italy

† Deceased.
\* lucca@marionegri.it

**Data Availability Statement:** Data cannot be shared publicly because no mention of the possibility of sharing the information, however

## Abstract

### Background

Mild anemia is a frequent although often overlooked finding in old age. Nevertheless, in recent years anemia has been linked to several adverse outcomes in the elderly population. Objective of the study was to investigate the association of mild anemia (hemoglobin concentrations: 10.0–11.9/12.9 g/dL in women/men) with all-cause mortality over 11–15 years and the effect of change in anemia status on mortality in young-old (65–84 years) and old-old (80+ years).

### Methods

The *Health and Anemia* and *Monzino 80-plus* are two door-to-door, prospective population-based studies that included residents aged 65-plus years in Biella municipality and 80-plus years in Varese province, Italy. No exclusion criteria were used.

### Results

Among 4,494 young-old and 1,842 old-old, mortality risk over 15/11 years was significantly higher in individuals with mild anemia compared with those without (young-old: fully-adjusted HR: 1.35, 95%CI, 1.15–1.58; old-old: fully-adjusted HR: 1.28, 95%CI, 1.14–1.44). Results were similar in the disease-free subpopulation (age, sex, education, smoking history, and alcohol consumption adjusted HR: 1.54, 95%CI, 1.02–2.34). Both age groups showed a dose-response relationship between anemia severity and mortality (*P* for trend <0.0001). Mortality risk was significantly associated with chronic disease and chronic kidney disease mild anemia in both age groups, and with vitamin B$_{12}$/folate deficiency and unexplained mild anemia in young-old. In participants with two hemoglobin determinations,

non-identifiable, collected during the survey with that of other research groups was reported in the informed consents signed by subjects (most of whom have already died) and informants participating in the studies. Requests should be addressed to the local ethics committee: Azienda Ospedaliera Maggiore della Carità di Novara (segreteria.scientifica@comitatoeticonovara.it) and Azienda Sanitaria Locale (A.S.L.) della Provincia di Varese (comitato.etico@asst-settelaghi.it).

**Funding:** This study was supported by Cariplo Foundation, Milan, Italy (Project # 2016-0860) and Italian Ministry of Health, Italy (Ricerca Finalizzata 2016 – Project # RF 2016-02364918). The Monzino 80-plus Study is being supported by a research grant from Italo Monzino Foundation, Milano, Italy. Present and past funders of the studies had no role in the design and conduct of the study; collection, management, analysis, and interpretation of data; preparation, review, or approval of the manuscript; or in the decision to submit the manuscript for publication.

**Competing interests:** The authors have declared that no competing interests exist.

seven-year mortality risk was significantly higher in incident and persistent anemic cases compared to constant non-anemic individuals in both age groups. In participants without anemia at baseline also hemoglobin decline was significantly associated with an increased mortality risk over seven years in both young-old and old-old. Limited to the Monzino 80-plus study, the association remained significant also when the risk was further adjusted also for time-varying covariates and time-varying anemia status over time.

## Conclusions

Findings from these two large prospective population-based studies consistently suggest an independent, long-term impact of mild anemia on survival at older ages.

## Introduction

From around midlife, hemoglobin concentration tends to decline and prevalence and incidence of anemia to increase with age [1,2]. It has been estimated that some 164 million elderly persons worldwide are affected by anemia [3], a number destined to sharply rise in coming decades owing to population aging throughout the world [4]. Mild anemia is a common laboratory finding in the elderly population, often perceived as having no clinical relevance, as a mere consequence of aging, or as a marker of chronic disease having no independent detrimental effect [1,5–10]. Conversely, in recent years all-grade anemia has been repeatedly linked to a number of adverse outcomes such as worse cognitive function, mood and quality of life, as well as increased disability, morbidity, and hospitalization [7]. Anemia has also been associated with major health indicators such as increased risk of mortality in elderly populations [7]. However, evidence from long-term population-based studies is limited and only one study has investigated the effect of anemia in a narrow age-range of oldest-old [11,12], the fastest-growing segment of the population and that with the highest prevalence and incidence of anemia [1]. Furthermore, few population-based studies have examined mild anemia impact on mortality, mostly comparing specific intervals of hemoglobin concentration with variable reference categories [11–17]. Except for two studies on all-grade anemia [17,18], no population-based study other than the *Health and Anemia Study* (*H&A*) [16] has so far explored the association of mild anemia types with the risk of mortality in the elderly population. Almost all studies assessed the relationship between anemia and mortality exclusively in prevalent cases, whose disorder however developed before the study started and thus were already exposed to the condition at the time of blood sampling. Moreover, a single measurement of hemoglobin concentration cannot investigate the association of change in anemia/non-anemia status to subsequent mortality.

Objective of the present study was to prospectively investigate the long-term association of mild anemia and mild anemia types on all-cause mortality in the young-old (65–84 years) and old-old (80+ years) from two population-based studies. Previous analyses in a cohort of young-old from *H&A65-84* have been extended from 3.5 to 15 years of observation. To examine the association of mild anemia with mortality in the old-old, the older cohorts from the *H&A* (*H&A85+*) and the *Monzino 80-plus* (*M80+*) studies, were pooled and followed-up over a period of 11 years (for the *M80+*, mortality data were available for a 15 year period). Finally, to assess the effect of change in anemia status over time on mortality, data from participants in *H&A65-84* and *M80+* with at least two hemoglobin determinations were analyzed.

## Methods

### Study settings and participants

*H&A* is a prospective, door-to-door population-based study of 65 years and older residents in the municipality of Biella, Italy. All registered individuals aged 65 to 84 years in 2003 were eligible (study years: 2003–2018). In 2007 the study was extended to all residents 85 years or older (study years: 2007–2018). To assess change in anemia status, a stratified random sample of individuals from *H&A65-84* was recontacted during 2005–2006 [1]: initial participants were stratified at baseline in anemic and non-anemic strata, all eligible consenting anemic and a random sample of eligible consenting non-anemic participants were included in the study on the association of mild anemia with cognitive, functional, mood, and QoL outcomes [21]. In *H&A65-84*, a second blood sample was available for 692 subjects (baseline: mean age 73.2 years, 55.4% women, mean [SD] hemoglobin concentration 13.6 [1.7] g/dL, 24.4% anemic): among the 344 consenting participants with baseline anemia, 29 died and 146 were not found or withdrew their consent to participate at the time of second blood draw; among the 655 consenting participants without anemia at baseline, 20 died and 112 were non traceable or withdrew their consent to participate at the time of the second blood draw. Baseline characteristics of participants with one (3,802) and two (692) samplings were for the most comparable. The mean (SD) time between samplings was 2.2 (0.1) years (median follow-up: 7.0 years; person-years of observation: 4,265; mortality rate: 3.8 per 100 person-years).

The *M80+* is a prospective, door-to-door, population-based study among oldest-old registered residents in the province of Varese, Italy (study years: 2002–2017) aimed at investigating cognitive decline and dementia. Very old ages were deliberately overrepresented in *M80+* in order to investigate the relation between cognitive function and age also in centenarians: to increase the number of very old, the survey was extended from the initial eight neighboring municipalities to all registered individuals aged 100 or older and to a random sample of men aged 95–99 residing in the remaining municipalities of Varese province in 2009–2010. Participants who had consented to donate a blood sample were asked at following visits whether they would agree to further blood sampling [19]. Of the initial 1,115 participants, 366 consented to donate a second blood sample (mean age: 89.7; men: 23.0%; with anemia: 23.8%; mean Hb: 13.1 g/dL); 667 did not or could not donate a second blood sample (mean age: 90.4; men: 26.5%; with anemia: 36.0%; mean Hb: 12.7 g/dL) and 82 died before the next visit (mean age: 92; men: 28.1%; with anemia: 57.3%; mean Hb: 11.8 g/dL). Of the 667 oldest-old with only one blood sample, 620 continued to participate in the *M80-plus* study but refused a second blood sample; 30 could not be traced and 17 refused to continue to participate in the *M80-plus* study. The mean (SD) time between samplings was 1.7 (1.0) years (median follow-up: 3.1 years; person-years of observation: 1,298; mortality rate: 22.4 per 100 person-years).

No exclusion criteria were used in either study. Lists of residents were obtained from the municipality registry offices. Residents living in nursing homes were included. Full details of study populations and design have already been published [1,16,19,20]. Dates of deaths were obtained from Municipal Registry Offices and Local Health Authorities.

Study protocols were approved by local research ethics committees: Hospital Health Authority of Novara and Azienda Sanitaria Locale (ASL) of Varese Province. Written informed consent was obtained from participants before blood sampling. Participants in the *M80+* study had also previously signed the informed consent to participate in the main study. Written informed consent was also obtained from informants in both studies. Study procedures were in accordance with the Helsinki Declaration of 1975, as revised in 2008.

## Study design

Venous blood samples were collected at the place of residence by registered nurses in the morning from participants in a sitting position.

Questionnaires were administered after blood sampling in the *H&A* study. In the *M80+* study, blood sampling was carried out in an *ad hoc* additional visit at the place of residence on average within 1.9 months of the scheduled study visit during which, after the administration of the questionnaire and tests, the subject's willingness to participate in the blood sub-study was investigated. A questionnaire was administered by specifically trained registered nurses in the *H&A* and psychologists (in *H&A85+*, *M80+*, and *H&A65-84* sample entered in the incident study) to ascertain habits, present and past diseases, and hospital admissions. Agreement between interviewers (nurses and psychologists) on medical history queries on a large sample of participants in the *H&A* study was very high (Cohen's k between 0.84 and 0.93) [21] and psychologists in the *H&A* employed the same questionnaire to collect the medical history used by the psychologists in the *M80+* study. Information on participants' health status was also obtained from the Epidemiology Unit of Biella ASL for *H&A* and from General Practitioners and nursing home geriatricians for *M80+*.

All participants entered the cohorts on the date of their blood draw and were followed until death or end of the study period. The follow-up period in which death from any cause could be ascertained was 15 years for *H&A65-84* and *M80+* cohorts and 11 years for *H&A85+* cohort. Death was the only censoring event. Dates of death between blood sampling and December 31, 2017 for *M80+* and June 30, 2018 for *H&A*, were obtained from Municipal Registry Offices and Local Health Authorities.

## Definitions of anemia and mild anemia

Hemoglobin concentration together with the other tests included in the complete blood count were determined on automated hematology analyzer instruments at the laboratory of Biella Hospital (*H&A*) and Laboratorio Milano, Milano (*M80+*). Anemia was defined according to the most commonly used WHO criteria (1968) as a hemoglobin concentration lower than 12 g/dL in women and 13 g/dL in men [22]. Along with most grading systems, mild anemia was defined by Dallman (1984), Groopman and Itri (1999), Wilson et al. (2004) as a hemoglobin concentration between 10.0 and 11.9 g/dL in women and 10.0 and 12.9 g/dL in men [23–25]. Anemia types were ascertained at baseline and their definitions are reported in S1 Methods.

## Statistical analysis

To investigate the association of anemia/mild anemia with mortality over 11 years in the old-old, individual participant data from the *H&A85+* and *M80+* cohorts were pooled. The rationale behind this pooling was that both studies were prospective door-to-door population-based studies in the old-old with no exclusion criteria other than age; both had a long lasting follow-up and were conducted during more or less matching calendar years; life expectancy was very similar; mortality data on an ongoing basis from the Municipal Registry Offices was available for both cohorts; the modalities to collect health-related information were very similar or identical in both. Kaplan-Meier curves were constructed for types of mild anemia and differences in survival were tested using log-rank tests. Age- and sex-corrected hazard ratios (HRs) and 95% confidence intervals (CIs) of death in the mildly anemic group compared to non-anemic group were calculated using Cox proportional hazard regression models. In the "fully"-adjusted model, HRs were further adjusted for the potential confounding effect of education, smoking history, alcohol consumption, previous year hospital admission, history of diabetes, hypertension, myocardial infarction, heart failure, chronic respiratory failure,

chronic renal insufficiency, cancer, transient ischemic attack, stroke, parkinsonism, dementia. All covariates were assessed at baseline and also at second sampling for participants with two blood samples. Limited to the Monzino 80-plus study, beyond baseline, other 8 follow-up assessments were available; it was thus possible to further adjust the multivariable model also for time-varying covariates (age, habits, and intervening comorbidities) in subjects with one (prevalent cases) as well as two hemoglobin determinations (incident cases and hemoglobin decline). Moreover, in participants with two hemoglobin determinations it was also possible to investigate how the association between anemia and mortality was affected by time-varying covariates together with time-varying exposure (anemia status). Overall, missing data on all covariates were only 0.5% in *H&A65-84* and 1.4% in the pooled 80+ cohort. To examine whether the effect of mild anemia was similar over time, two survival analyses were set up: from 0 to 7 years and, in all participants who survived the first seven years of follow up, from 7 (the start of the 8th year) to 15/11 years. To assess whether a dose-response relationship existed, hemoglobin concentrations were divided into categories of 1 mg/dL and p for trend analyses carried out. Subgroup analyses explored the potential risks of mortality associated with different types of anemia. Each type of anemia was separately compared with non-anemia. Proportionality assumption of the models was checked inspecting the log(-log(survival)) versus log (survival time) graphs: no departure from parallelism was found.

To investigate whether WHO criteria (1968) may have affected the estimated effect of mild anemia on mortality, we re-evaluated this association using slightly higher lower limits of normal hemoglobin concentration to define anemia in white adults proposed by Beutler and Waalen in 2006 (lower than 12.2 g/dL in women and lower than 13.2 g/dL in men) [26]. We further tested this association also using recent WHO criteria (2011) for mild anemia (lower limit of hemoglobin concentration: 11 g/dL) [27]. All p values were two tailed. Data were analyzed using JMP Pro v. 15.0 (SAS Institute Inc., Cary, NC, USA) and Stata v.15.1 (Stata Corp, College Station, Tx).

## Results

### Population characteristics

The two study populations comprised 6,336 subjects (S1 Fig). Of 10,092 living individuals contacted in *H&A*, 5,221 (51.7%) accepted blood tests: 4,494 aged 65–84 years and 727 85 years or more at blood sampling. Compared with non-participants, participants were less than 1 year younger (74.4 versus 73.5) and with a similar proportion of women (61.8% versus 60.2%). In *M80+*, 1,115 of 2,039 participants aged 80 years or older alive at first interview consented to donate a subsequent blood sample (54.7%). Mean age and proportion of women were similar between subjects who accepted blood sampling (89.7 years; 74.5%) and those who did not (90.5 years; 71.1%), and there was no significant difference between groups regarding primary research targets: dementia prevalence and cognitive function (age-, sex- and education-adjusted logistic regression analysis: p = 0.233 and p = 0.221).

Table 1A and 1B show main characteristics of study cohorts. Unsurprisingly, mean age was higher in *M80+* than in *H&A85+* because very old ages were deliberately overrepresented in *M80+* [20]. Individuals aged 85 years or older included in *H&A85+* and *M80+* cohorts were respectively born between 1902 and 1923, and 1900 and 1921. Life expectancy at age 85 was 6.184 years in Varese province (2002) and 6.117 in Biella province (2007) [28].

### Risk of mortality associated with mild anemia and mild anemia types

*H&A65-84* **cohort.**   During 15 years of follow-up after blood sampling, 230 anemic (66.9%; 205 mild anemic) and 1,928 non-anemic (46.5%) individuals died (Fig 1). The median

**Table 1.** A. Baseline characteristics of participants aged 65–84 years at blood sample from the *Health and Anemia 65–84* population-based study. B. Baseline characteristics of participants aged 80 years or older at blood sample from two pooled population-based studies (*Health and Anemia 85+* and *Monzino 80-plus*).

| | A | | | | B | | | |
|---|---|---|---|---|---|---|---|---|
| | No anemia | Mild anemia[a] | Moderate-severe anemia | All | No anemia | Mild anemia[a] | Moderate-severe anemia | All |
| Participants, No. | 4,150 | 313 | 31 | 4,494 | 1,283 | 478 | 81 | 1,842 |
| Female sex, No. (%) | 2,521 (60.6) | 169 (54.0) | 22 (71.0) | 2,706 (60.2) | 988 (77.0) | 310 (64.9) | 64 (79.0) | 1,362 (73.9) |
| Male sex, No. (%) | 1,635 (39.4) | 144 (46.0) | 9 (29.0) | 1,788 (39.8) | 295 (23.0) | 168 (35.2) | 17 (21.0) | 480 (26.1) |
| Age, mean (SD), years | 73.4 (5.2) | 75.1 (5.5) | 76.0 (5.4) | 73.5 (5.2) | 89.4 (4.7) | 91.2 (5.2) | 91.2 (6.3) | 90.0 (5.0) |
| Education, mean (SD), years | 7.8 (3.8) | 7.1 (3.9) | 7.3 (4.1) | 7.7 (3.8) | 5.7 (3.1) | 5.7 (3.2) | 5.0 (2.5) | 5.7 (3.2) |
| Current smokers, No. (%) | 618 (14.9) | 36 (11.5) | 2 (6.7) | 656 (14.6) | 42 (3.3) | 11 (2.3) | 0 | 53 (2.9) |
| Former smokers, No. (%) | 1,319 (31.9) | 98 (31.3) | 11 (36.7) | 1,428 (31.9) | 230 (18.2) | 100 (21.3) | 15 (19.5) | 345 (19.1) |
| Current alcohol use, No. (%) | 3,017 (73.2) | 212 (68.2) | 15 (50.0) | 3,244 (72.7) | 729 (57.7) | 237 (50.1) | 34 (43.6) | 1,000 (55.2) |
| Former alcohol use, No. (%) | 138 (3.4) | 22 (7.1) | 1 (3.3) | 161 (3.6) | 121 (9.6) | 76 (16.1) | 16 (20.5) | 213 (11.8) |
| Body mass index, mean (SD) | 25.0 (4.1) | 24.6 (4.2) | 25.2 (7.4) | 25.0 (4.1) | 23.5 (4.2) | 22.9 (4.2) | 23.6 (4.9) | 23.4 (4.3) |
| Diabetes, No. (%) | 364 (8.8) | 53 (17.1) | 4 (12.9) | 421 (9.4) | 165 (13.0) | 65 (13.8) | 17 (21.8) | 274 (13.6) |
| Hypertension, No. (%) | 2,206 (53.9) | 172 (56.2) | 17 (56.7) | 2,395 (54.1) | 809 (64.0) | 265 (56.1) | 44 (56.4) | 1,118 (61.6) |
| History of cancer, No. (%) | 462 (11.3) | 39 (12.9) | 10 (32.3) | 511 (11.5) | 181 (14.3) | 93 (19.7) | 16 (20.5) | 290 (16.0) |
| Hemoglobin level, g/dL, mean (SD) | 14.2 (1.2) | 11.6 (0.7) | 8.9 (1.1) | 14.0 (1.4) | 13.6 (1.0) | 11.5 (0.7) | 8.8 (1.0) | 12.9 (1.6) |
| Mean Corpuscular Volume, mean (SD) | 96.1 (6.3) | 87.5 (13.2) | 88.6 (14.2) | 95.4 (7.3) | 92.6 (5.4) | 91.3 (8.1) | 84.7 (13.6) | 91.9 (6.9) |

[a]Concentration of hemoglobin: 10.0–11.9 (women) or 10.0–12.9 (men) g/dL [23–25].

follow-up period was 14.0 years with 50,522 person-years of observation and a mortality rate of 4.3 per 100 person-years (4.1 in non-anemic and 7.1 in mild anemic participants). Over the 15-year follow-up, mortality risk was significantly increased in participants with anemia (fully-adjusted HR: 1.42; 95% CI, 1.22–1.65) and mild anemia (fully-adjusted HR: 1.35; 95% CI, 1.15–1.58). In the first seven years after blood collection (median follow-up: 7.0 years; person-years of observation: 28,607 years; mortality rate: 2.9 per 100 person-years), compared with non-anemic, mortality risks were significantly higher in individuals with anemia (fully-adjusted HR: 1.72; 95% CI, 1.39–2.12) and mild anemia (fully-adjusted HR: 1.60; 95% CI, 1.28–2.00) (Table 2). From 7 to 15 years (3,574 participants; median follow-up: 7.4 years; person-years of observation: 21,915; mortality rate: 6.1 per 100 person-years) the risk was still increased in anemic young-old but the difference between groups was no longer significant (Table 2). Fig 2 shows the relationship between categorical hemoglobin concentrations and 7-year mortality risk: a dose-response relationship between anemia severity and mortality was observed (*P* for trend < 0.0001) and fully-adjusted HRs of death were higher for hemoglobin concentrations below WHO criteria.

Compared with subjects without anemia, 7-year mortality risk was significantly increased in subjects with normocytic (MCV 80–100 fL: fully-adjusted HR: 1.66, 95% CI, 1.26–2.19) and macrocytic (MCV>100 fL: fully-adjusted HR: 2.14; 95% CI, 1.49–3.09) mild anemia. To further investigate the association between mild anemia and mortality, we performed a subgroup analysis by anemia type. Over the first seven years of follow-up, except for thalassemia trait, results for the other anemia types tended, in various degrees, to confirm the findings in the

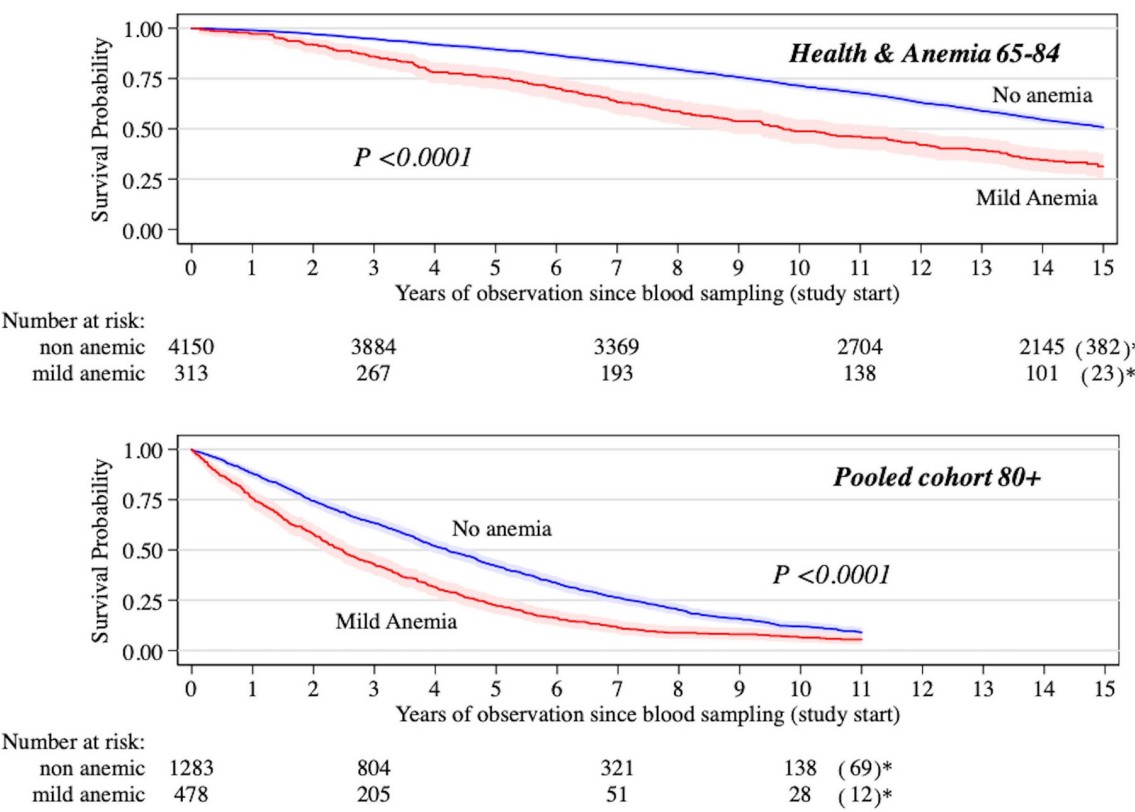

**Fig 1. Survival by mild anemia status in the *Health & Anemia 65–84* cohort and in the pooled *Health & Anemia 85+* and *Monzino 80+* cohort.** Kaplan-Meier curves for individuals aged 65–84 years and 80 years or older. The shaded area indicates 95% confidence intervals. Mild anemia was defined as hemoglobin concentration between 10.0 and 11.9 g/dl in women and between 10.0 and 12.9 g/dl in men [23–25]. *Individuals included in the studies first.

overall population with mild anemia: mortality risk was significantly increased in subjects with mild anemia due to chronic disease (fully-adjusted HR: 3.17; 95% CI, 2.10–4.78), chronic kidney disease (fully-adjusted HR: 2.65; 95% CI, 1.17–5.99), folate or vitamin $B_{12}$ deficiency (fully-adjusted HR: 2.12; 95% CI, 1.25–3.60), and unexplained causes (fully-adjusted HR: 1.51; 95% CI, 1.02–2.23) (Table 3, Fig 3).

*H&A85+ and M80+ cohorts.* During 11-year follow-up period, 526 anemic (94.1%; 449 mild anemic) and 1,137 non-anemic individuals (88.6%) died in the pooled 80+ cohort (Fig 1). The median follow-up period was 3.5 years with 7,871 person-years of observation and a mortality rate of 21.1 per 100 person-years (18.7 in non-anemic and 28.6 in mild anemic participants). Over the 11-year follow-up, mortality risk was significantly higher in participants with anemia (fully-adjusted HR: 1.32; 95% CI, 1.18–1.48) and mild anemia (fully-adjusted HR: 1.28; 95% CI, 1.14–1.44). In the first seven years after blood collection (median follow-up: 3.5 years; person-years of observation: 7,235 years; mortality rate: 19.7 per 100 person-years), compared with non-anemic, mortality risks were significantly higher in individuals with anemia (fully-adjusted HR: 1.42; 95% CI, 1.26–1.60) and mild anemia (fully-adjusted HR: 1.38; 95% CI, 1.21–1.56) (Table 2). From 7 to 11 years (287 participants; median follow-up: 2.3 years; person-years of observation: 637; mortality rate: 23.9 per 100 person-years) no significant difference in mortality risk between anemic and non-anemic old-old was found (Table 2). Limited to the *M80+* cohort, mortality risks associated with prevalent anemia and mild anemia were slightly increased when the model was further adjusted for time-varying covariates (anemia:

**Table 2. Risk of mortality in anemic and mild anemic compared with non-anemic participants aged 65–84 years at blood sample from the Health and Anemia 65–84 population-based study and participants aged 80 years or older at blood sample from two pooled population-based studies (Health and Anemia 85+ and Monzino 80-plus).**

| Anemia definitions | Model | Health & Anemia 65–84 (N = 4494) | | | Health & Anemia 85+ and Monzino 80+ (N = 1842) | | |
|---|---|---|---|---|---|---|---|
| | | 0–15 years | 0–7 years | 7–15 years | 0–11 years | 0–7 years | 7–11 years |
| Anemia: [Hb] g/dL | | Hazard ratios (95% confidence intervals) | | | Hazard ratios (95% confidence intervals) | | |
| ≤11.9 (W) or ≤12.9 (M)[a] | AS-A | 1.70 (1.48–1.95) | 2.13 (1.76–2.57) | 1.36 (1.10–1.66) | 1.43 (1.29–1.59) | 1.54 (1.38–1.73) | 0.76 (0.53–1.10) |
| | F-A | 1.42 (1.22–1.65) | 1.72 (1.39–2.12) | 1.21 (0.97–1.51) | 1.32 (1.18–1.48) | 1.42 (1.26–1.60) | 0.68 (0.44–1.03) |
| ≤12.1 (W) or ≤13.1 (M)[b] | AS-A | 1.48 (1.31–1.67) | 1.85 (1.55–2.20) | 1.22 (1.03–1.46) | 1.43 (1.29–1.58) | 1.51 (1.35–1.68) | 0.93 (0.68–1.28) |
| | F-A | 1.31 (1.15–1.50) | 1.59 (1.31–1.92) | 1.13 (0.93–1.36) | 1.28 (1.14–1.42) | 1.34 (1.20–1.51) | 0.87 (0.62–1.23) |
| Mild anemia: [Hb] g/dL | | Hazard ratios (95% confidence intervals) | | | Hazard ratios (95% confidence intervals) | | |
| 10.0[c]-11.9 (W) or 10.0–12.9 (M) | AS-A | 1.60 (1.38–1.85) | 1.98 (1.62–2.42) | 1.30 (1.05–1.61) | 1.39 (1.24–1.55) | 1.50 (1.33–1.69) | 0.69 (0.46–1.04) |
| | F-A | 1.35 (1.15–1.58) | 1.60 (1.28–2.00) | 1.18 (0.94–1.48) | 1.28 (1.14–1.44) | 1.38 (1.21–1.56) | 0.61 (0.39–0.98) |
| 11.0[d]-11.9 (W) or 11.0–12.9 (M) | AS-A | 1.53 (1.31–1.79) | 1.95 (1.57–2.41) | 1.21 (0.96–1.53) | 1.27 (1.12–1.44) | 1.38 (1.21–1.57) | 0.61 (0.38–0.96) |
| | F-A | 1.28 (1.07–1.52) | 1.56 (1.23–1.99) | 1.08 (0.83–1.39) | 1.22 (1.07–1.39) | 1.31 (1.14–1.50) | 0.60 (0.36–1.00) |
| 10.0[c]-12.1 (W) or 10.0–13.1 (M) | AS-A | 1.48 (1.30–1.69) | 1.81 (1.50–2.17) | 1.25 (1.04–1.50) | 1.39 (1.25–1.54) | 1.47 (1.31–1.65) | 0.90 (0.65–1.25) |
| | F-A | 1.29 (1.12–1.49) | 1.53 (1.25–1.87) | 1.14 (0.93–1.39) | 1.24 (1.11–1.39) | 1.30 (1.15–1.47) | 0.84 (0.58–1.20) |
| 11.0[d]-12.1 (W) or 11.0–13.1 (M) | AS-A | 1.42 (1.24–1.63) | 1.76 (1.45–2.14) | 1.18 (0.97–1.44) | 1.29 (1.15–1.44) | 1.36 (1.20–1.54) | 0.85 (0.60–1.21) |
| | F-A | 1.24 (1.07–1.44) | 1.49 (1.20–1.85) | 1.07 (0.86–1.32) | 1.18 (1.04–1.33) | 1.23 (1.08–1.40) | 0.86 (0.59–1.27) |

[Hb]: concentration of hemoglobin; W: women; M: men; AS-A: age- and sex-adjusted; F-A: "fully"-adjusted for baseline age, sex, education, smoking status, alcohol consumption, hypertension, diabetes, heart failure, myocardial infarction, chronic respiratory failure, chronic renal insufficiency, cancer, transient ischemic attack, stroke, parkinsonism, dementia, hospitalization during the previous year, and study (only for the two pooled studies).

[a]WHO criteria (1968) [22].

[b]Beutler and Waalen criteria (2006) for white adults [26].

[c]Dallman (1984); Groopman and Itri (1999); Wilson et al. (2004) [23–25].

[d]WHO criteria (2011) [27].

HR 1.59 [95% CI: 1.39–1.82] over 11 years and HR 1.64 [95% CI: 1.42–1.89] over seven years; mild anemia: HR1.57 (95% CI: 1.36–1.80) over 11 years and HR 1.60 (95% CI: 1.38–1.85) over seven years.

Relationship between categorical hemoglobin concentrations and 7-year mortality risk in the pooled 80+ cohort was analogous to that in the 65–84 cohort, with higher risk of death for hemoglobin concentrations below WHO criteria and a dose-response relationship between anemia severity and mortality (P for trend < 0.0001). (Fig 2).

Also in the old-old 7-year mortality risk was significantly increased only in subjects with normocytic (fully-adjusted HR: 1.38, 95% CI, 1.20–1.58) and macrocytic (fully-adjusted HR: 1.37, 95% CI, 1.01–1.87) mild anemia. In the old-old included in the *H&A85+* cohort, risk of mortality over 7 years was significantly higher in subjects with mild anemia of chronic disease (fully-adjusted HR: 1.82; 95% CI, 1.08–3.08) and chronic kidney disease (fully-adjusted HR: 1.62; 95% CI, 1.11–2.36) (Table 3, Fig 3).

**H&A65+ and M80+ pooled cohorts.** S1 Table summarizes the results in the pooled young-old and old-old cohorts. Compared with non-anemic, mortality risk was significantly higher in elderly individuals with mild anemia in the first seven years after blood collection (fully-adjusted HR: 1.40; 95% CI, 1.26–1.57) as well as over the entire 11-year follow-up period (fully-adjusted HR: 1.29; 95% CI, 1.17–1.43). From 7 to 11 years no significant difference in mortality risk between anemic and non-anemic elderly persons was found.

**Sensitivity analyses for the association between anemia or mild anemia and mortality.** Similar results were found using different definitions of anemia or mild anemia (Table 2). Risk

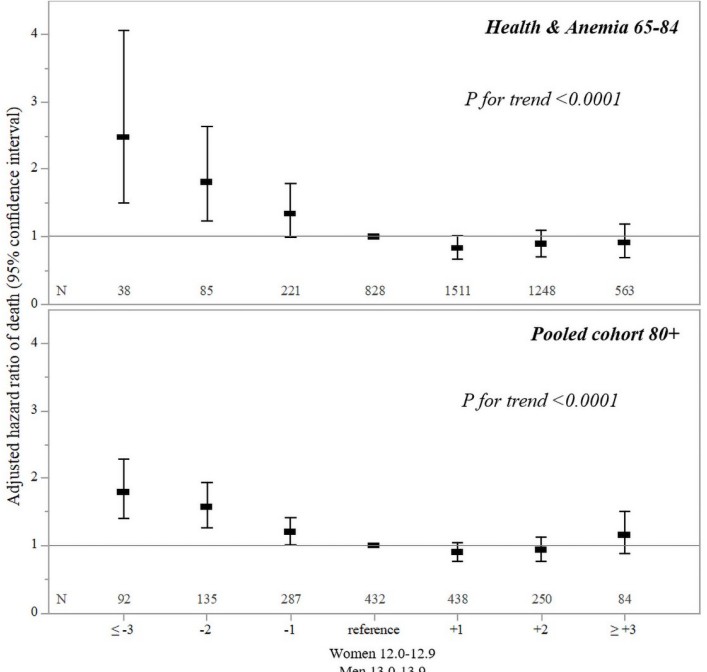

**Fig 2. Association between categorical hemoglobin concentration and 7-year mortality risk in the *Health & Anemia 65–84* cohort and in the pooled *Health & Anemia 85+* and *Monzino 80+* cohort.** Hazard ratios adjusted for age, sex, education, smoking status, alcohol use, hospitalization (previous year), hypertension, diabetes, heart failure, myocardial infraction, TIA, stroke, chronic respiratory failure, chronic renal insufficiency, cancer, parkinsonism, dementia, study (only for pooled cohort 80+).

**Table 3. Risk of mortality associated with mild anemia type over seven years in participants aged 65–84 and 85 or older years at blood sample from the *Health and Anemia* population-based study.**

| Mild anemia[a] type | Health & Anemia 65–84 | | | Health & Anemia 85+ | | |
|---|---|---|---|---|---|---|
| | Deceased/Total (%) | HR (95%CI) [b] | HR (95%CI) [c] | Deceased/Total (%) | HR (95%CI) [b] | HR (95%CI) [c] |
| No Anemia | 690/4150 (16.6) | 1.00 | 1.00 | 368/537 (68.5) | 1.00 | 1.00 |
| Thalassemia Trait | 8/62 (12.9) | 0.85 (0.42–1.70) | 0.75 (0.36–1.59) | 5/7 (71.4) | 1.25 (0.52–3.02) | 1.59 (0.59–4.32) |
| Iron Deficiency | 10/39 (25.6) | 1.74 (0.93–3.25) | 1.52 (0.79–2.96) | 15/17 (88.2) | 1.54 (0.91–2.59) | 1.32 (0.74–2.35) |
| Vitamin $B_{12}$ or Folate Deficiency | 15/31 (48.8) | 2.17(1.30–3.64) | 2.12 (1.25–3.60) | 9/13 (69.2) | 0.93 (0.48–1.80) | 1.17 (0.59–2.32) |
| Chronic kidney disease | 9/16 (56.3) | 3.34 (1.73–6.46) | 2.65 (1.17–5.99) | 39/42 (92.9) | 1.67 (1.19–2.33) | 1.62 (1.11–2.36) |
| Chronic Disease | 32/56 (57.1) | 4.27 (2.99–6.10) | 3.17 (2.10–4.78) | 22/24 (91.7) | 2.36 (1.53–3.65) | 1.82 (1.08–3.08) |
| Unexplained | 30/89 (33.7) | 1.66 (1.15–2.40) | 1.51 (1.02–2.23) | 25/35 (71.4) | 0.89 (0.59–1.34) | 1.04 (0.67–1.61) |

Each type of anemia was separately compared with non-anemia.

HR: hazard ratio; CI: confidence interval.

[a]Mild anemia: 10.0–11.9 (women) or 10.0–12.9 (men) g/dL [23–25].

[b]Age- and sex-adjusted.

[c]Adjusted for baseline age, sex, education, smoking status, alcohol consumption, hypertension, diabetes, heart failure, myocardial infarction, chronic respiratory failure, chronic renal insufficiency (except for anemia of chronic kidney disease), cancer (except for anemia of chronic disease), transient ischemic attack, stroke, parkinsonism, dementia, hospitalization during the previous year.

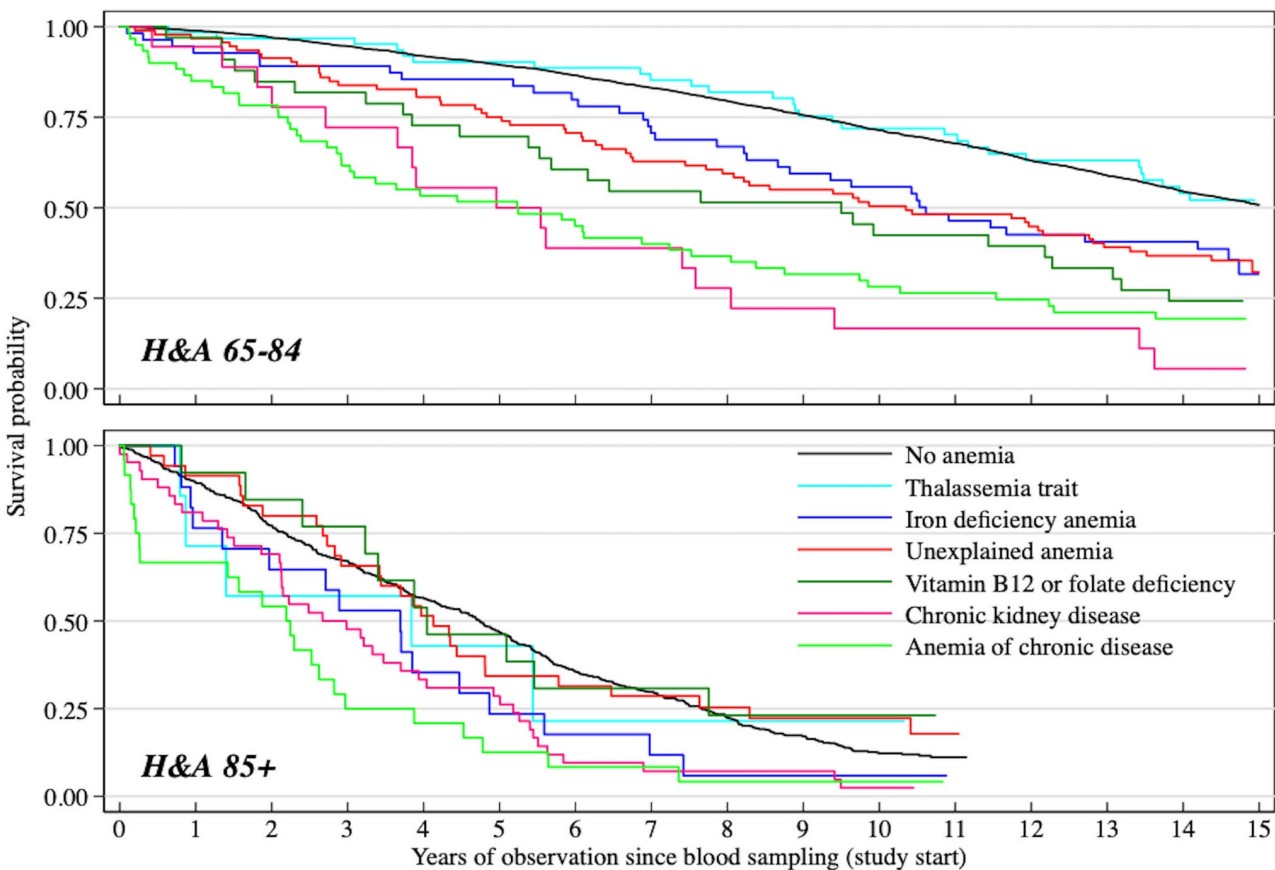

**Fig 3. Survival by mild anemia type in the *Health and Anemia (H&A)* population-based study.** Kaplan-Meier curves for individuals aged 65–84 years *(H&A65-84)* and 85 years or older *(H&A85+)*. Mild anemia was defined as hemoglobin concentration between 10.0 and 11.9 g/dl in women and between 10.0 and 12.9 g/dl in men [20, 22].

of mortality associated with anemia and mild anemia in subjects 85 years and older in H&A85+ and M85+ cohorts separately were quite similar (S2 Table). If only subjects aged 85+ from the two cohorts were pooled, results were comparable (S3 Table). Results were akin also setting-up different cohorts: *H&A65-79* (S4 Table), *H&A80+* together with the *M80+* (that is all individuals aged 80 years or older; S4 Table), *H&A65+* (S5 Table), or *M80+* (S5 Table). Risks over a follow-up period of 15 or 11 years in the *M80+* were more or less overlapping (S6 Table). Almost identical results were also found when body mass index was added to the fully-adjusted model or when glomerular filtration rate estimated with MDRD formula was included instead of the categorical renal insufficiency variable (S7 Table). When a co-morbid disease severity index (see S1 Methods) was added to the fully-adjusted model, the association of mild anemia with mortality over 7 years remained almost unchanged both in *H&A65-84* (HR: 1.63; 95% CI, 1.31–2.03) and in *H&A85+* (HR: 1.36; 95% CI, 1.10–1.68). If the 63 participants with thalassemia trait (62 with mild and 1 with moderate anemia) were removed from the analysis of the *H&A65-84* cohort, hazards of death from mild anemia would consistently increase with respect to those analyzing the complete cohort: fully-adjusted HRs (95% CIs): 1.45 (1.22–1.72) over 0–15 years, 1.78 (1.41–2.25) over 0–7 years, and 1.21 (0.94–1.57) over 7–15 years.

**Risk of mortality associated with mild anemia in "healthy" elderly subjects.** At baseline 1,093 elderly persons (mean age 74.6, 64% women) from both H&A65+ and M80+ had no history of any of the diseases entered as confounders in multivariable analyses. The median

**Table 4. Risk of mortality associated with incident anemia, decline in hemoglobin concentration, and change in anemia status in young-old and old-old participants with two blood samples over 7 years after the second blood sampling.**

| Anemia status overtime | Health & Anemia 65–84 | | Monzino 80-plus | |
|---|---|---|---|---|
| | N cases/N All | F-A HR (95% CI) | N cases/N All | F-A HR (95% CI) |
| Incident anemia | 27/523[a] | 2.81 (1.47–5.38)[b] | 65/279[a] | 1.47 (1.04–2.07)[b] |
| [Hemoglobin] decline | 523[a] | 1.39 (1.10–1.76)[c] | 279[a] | 1.22 (1.08–1.39)[c] |
| No anemia→Anemia | 27/692[d] | 2.48 (1.33–4.64)[b] | 65/366[d] | 1.53 (1.09–2.13)[b] |
| Anemia→Anemia | 118/692[d] | 1.75 (1.13–2.70)[b] | 61/366[d] | 1.57 (1.10–2.25)[b] |
| Anemia→No anemia | 51/692[d] | 1.59 (0.86–2.97)[b] | 26/366[d] | 1.32 (0.80–2.17)[b] |

HR: hazard ratio; CI: confidence interval; F-A "fully"-adjusted for age, sex, education, smoking, alcohol, hypertension, diabetes, heart failure, myocardial infarction, chronic respiratory failure, chronic renal insufficiency, TIA, stroke, cancer, dementia, hospitalization during the previous year. All covariates were assessed at second sampling.

[a]Subjects without anemia at first of two samplings.

[b]Reference group: subjects consistently non-anemic at both samplings.

[c]Per 1 g/dL decrease of hemoglobin concentrations. Furtherly adjusted also for hemoglobin concentration at first sampling.

[b]Subjects with two samplings.

follow-up period was 7 years with 7,043 person-years of observation and a mortality rate of 2.5 per 100 person-years (2.2 in non-anemic and 7.6 in mild anemic participants). Individuals with mild anemia (N = 74) showed an increased risk of mortality compared to those without over the first seven years after blood collection (baseline age, sex, education, smoking history, and alcohol consumption adjusted HR: 1.54, 95% CI, 1.02–2.34).

**Risk of mortality associated with anemia status and change in hemoglobin concentration assessed over time.** Considering the 523 subjects without anemia at first sampling in *H&A65-84* (median follow-up: 7.0 years; person-years of observation: 3,286; mortality rate: 3.3 per 100 person-years), the 27 incident cases at second sampling (5.2%; mortality rate: 11.4 per 100 person-years) showed an increased risk of mortality over the 7 years following the second blood sample compared to those who did not develop anemia (fully-adjusted HR: 2.81; 95% CI, 1.47–5.38). In this cohort of young-old without anemia, also hemoglobin decline over the same period after the second sampling was associated with an increased risk of mortality: fully-adjusted (also for hemoglobin at first sampling) HR: 1.39 (95%CI, 1.10–1.76) per 1 g/dL decrease of hemoglobin concentration (Table 4). When subjects were categorized according to anemia status at both samplings (Table 4), compared to subjects constantly non-anemic (n = 496), a significantly increased risk of mortality over 7 years after the second sampling was found in those with persistent anemia (n = 118, fully-adjusted HR: 1.75; 95% CI, 1.13–2.70) and in those who became anemic at second sampling (n = 27, fully-adjusted HR: 2.48; 95% CI, 1.33–4.64). In anemic subjects at first sampling who were non-anemic at second sampling (n = 51), the association was not significant (fully-adjusted HR: 1.59; 95% CI, 0.86–2.97).

Considering the 279 subjects without anemia at first sampling in *M80+* (median follow-up: 3.4 years; person-years of observation: 1,048; mortality rate: 20.5 per 100 person-years), the 65 incident cases (23.3%) at second sampling (23.3%; mortality rate: 31.8 per 100 person-years) showed an increased risk of mortality over the 7 years following the second blood sample compared to those who did not develop anemia (fully-adjusted HR: 1.47; 95% CI, 1.04–2.07). In this cohort of old-old without anemia, also hemoglobin decline over the same period after the second sampling was associated with an increased risk of mortality: fully-adjusted (also for hemoglobin level at first sampling) HR: 1.22 (95%CI, 1.08–1.39) per 1 g/dL decrease of hemoglobin concentration (Table 4). When subjects were categorized according to anemia status at both samplings (Table 4), compared to subjects constantly non-anemic (n = 214), a

**Table 5. Risk of mortality associated with prevalent anemia, incident anemia, and decline in hemoglobin concentration in old-old participants in the Monzino 80-plus study.**

| | N cases/N All | Model | Hazard ratios (95% confidence intervals) | | |
|---|---|---|---|---|---|
| | | | **0–11 years** | **0-7years** | **7–11 years** |
| Prevalent Anemia | 374/1,115 | AS-A | 1.62 (1.42–1.85)[a] | 1.70 (1.48–1.95)[a] | 0.90 (0.41–1.97)[a] |
| | | F-A | 1.59 (1.39–1.82)[a] | 1.64 (1.42–1.89)[a] | 0.91 (0.38–2.20)[a] |
| Prevalent Mild Anemia | 323/1,064 | AS-A | 1.60 (1.39–1.83)[a] | 1.67 (1.45–1.93)[a] | 0.82 (0.34–1.99)[a] |
| | | F-A | 1.57 (1.36–1.80)[a] | 1.60 (1.38–1.85)[a] | 0.98 (0.37–2.54)[a] |
| Subjects with 2 blood samplings | 366[b] | AS-A | 3.33 (2.02–5.50)[a] | 2.72 (1.44–5.15)[a] | 1.38 (0.24–7.95)[a] |
| | | F-A[c] | 2.88 (1.73–4.80)[a] | 2.29 (1.19–4.40)[a] | 2.67 (0.42–17.0)[a] |
| Incident anemia | 65/279[d] | AS-A | | 1.67 (1.23–2.27)[e,f] | |
| | | F-A | | 1.61 (1.17–2.22)[e,f] | |
| [Hemoglobin] decline in non-anemic subjects | 279[d] | AS-A | | 1.26 (1.13–1.41)[e,g] | |
| | | F-A | | 1.27 (1.14–1.42)[e,g] | |

AS-A: age- and sex-adjusted; F-A: "fully"-adjusted for time-varying covariates: age, sex, education, smoking, alcohol, hypertension, diabetes, heart failure, myocardial infarction, chronic respiratory failure, chronic renal insufficiency, TIA, stroke, cancer, dementia, hospitalization during the previous year.

[a]Compared with participants without anemia at first blood sampling.

[b]Subjects with two blood samplings.

[c]Furthermore adjusted also for time-varying covariates and anemia status.

[d]Subjects without anemia at first of two blood samplings.

[e]0-7 years of follow-up starting from the second sampling for "incident anemia" and "[Hemoglobin] decline in incident cases".

[f]Reference group: subjects consistently non-anemic at both samplings for incident cases.

[g]Per 1 g/dL decrease of hemoglobin concentrations. Hazard ratios furthermore adjusted also for hemoglobin concentration at first sampling.

significantly increased risk of mortality over 7 years after the second sampling was found in those who were consistently anemic (n = 61, fully-adjusted HR: 1.57; 95% CI, 1.10–2.25) and in those who became anemic at second sampling (n = 65, fully-adjusted HR: 1.53; 95% CI, 1.09–2.13). In anemic at first sampling who were non-anemic at second sampling (n = 26), the association was not significant (fully-adjusted HR: 1.32; 95% CI, 0.80–2.17). In all participants in the *M80+* with two hemoglobin determinations it was possible to control for both time-varying anemia status and time-varying covariates: the risk associated with anemia resulted moderately increased also with respect to the model further adjusted only for time-varying covariates (Table 5). In non-anemic subjects with two hemoglobin determinations, the risk of mortality associated with incident anemia or change in hemoglobin concentration over the seven years following the second sampling was increased when also time-varying covariates were added to the "fully"-adjusted model (Table 5).

## Discussion

In these two large prospective population-based studies, elderly persons with mild anemia had an overall 40% increased risk of dying, 60% in young-old and 38% in old-old. Although gradually attenuated over time, the effect of mild anemia on survival was still detectable after 15/11 years. The risk was independent of multifarious potential confounders including comorbid conditions and disease severity, and remained significantly increased also in the disease-free subpopulation, as in a study on anemia of any severity [14]. Results remained stable also when different definitions of anemia and mild anemia were applied. A dose-response relationship between anemia severity and mortality was found in both elderly age groups, while, in agreement with other population-based studies [12,13 (in men),14,17], we did not find a significantly increased mortality risk for hemoglobin concentrations higher than WHO cut-offs for anemia. Most prevalent cases remained persistently anemic over time and their risk of

mortality had similar estimates to that in anemic subjects who had had only one sampling. Consistent with two studies in a selected population of young-old [29] and 85 years old [11], incident anemia and decline in hemoglobin concentration were ~~was~~ significantly associated with increased risk of mortality in both age cohorts. In the Monzino 80-plus study the risk associated with prevalent and incident anemia was even higher when the multivariable model could be further adjusted also for time-varying covariates and, limited to all old-old with two blood samplings, for change in anemia status at second sampling. The risk of mortality was associated with mild anemia due to chronic disease and chronic kidney disease in both young-old and old-old. Also unexplained and vitamin $B_{12}$ or folate deficiency mild anemias were linked to an increased mortality risk, but reached significance only in the young-old. Similar results have been observed in young-old with anemia of any severity [17,18,30].

Taken together, all these remarkably consistent findings in two population-based studies strongly support an independent and enduring impact of mild anemia on survival in both young-old and old-old and expands previous results on anemia of any severity as a predictor of mortality almost exclusively in the young-old over shorter periods of time in population-based [12–17,31] and selected population [32–34 (in whites),35–37] studies.

Multiple factors are likely to contribute to mild anemia onset in many elderly cases [1,7]. The interaction between low hemoglobin concentration and underlying physio-pathological processes is complex. Although mild anemia could partially be a marker of systemic physical decline or a concomitant of chronic diseases [7–9], its steady increase in prevalence and incidence with age [1,2] as well as the evidence that the risk of mortality associated with mild anemia is very similar in the overall population and in the healthy subpopulation, seem to indicate the presence of specific age-related changes. However, the precise mechanisms by which mild anemia has an independent detrimental effect on relevant health-related outcomes in older persons are not fully understood [7]. Anemia of chronic disease, which accounts for about one third of the anemia cases in the elderly population [1,2], is a condition associated with chronic immune activation [38]. Nevertheless, whether this chronic inflammatory condition reflects a primary immune dysregulation associated with the aging process, develops as a systemic response to age-related co-morbidities, or arises from the interaction between these two mechanisms is unclear [7,8,38]. The etiology of about one fourth of mild anemia cases in older age remains unexplained and a portion of them might be accounted for by myelodysplastic syndromes [1,2,7,8,10,16,39,40]. Increased mortality in incident cases together with no evidence of mild anemia impact on survival of individuals with a genetic, usually asymptomatic, lifetime condition such as thalassemia trait, seem to suggest that when developed at older age, mild anemia could significantly add to health vulnerability and poor clinical outcomes.

The present study has several strengths, primarily the possibility to investigate the relationship between mild anemia and mortality in two different prospective population-based studies. Notwithstanding their advanced age, cohorts were followed-up over a very long time. Repeated measures of hemoglobin concentration permitted controlling for the effect of changing anemia status over time and assessing the risk of mortality following the incidence of anemia and decline in hemoglobin concentration. The association with mortality was investigated using several suggested definitions of anemia and mild anemia with consistent results. Analyses were adjusted for many potential confounders and replicated in disease-free subjects. No previous study has attempted to adjust also for the influence of change in anemia/non-anemia status and time-varying covariates. Classification of anemia types and the presence of a sufficiently large group of young-old with an inherited mild anemia have permitted a more thorough examination of the link between mild anemia and mortality in the elderly population.

The study also has potential limitations. Although prospective cohort studies can help to assess a causal association, further experimental trials would contribute to establish true

causality. Considering the moderate response rate, the possible influence of a non-response bias on the association investigated cannot be excluded. However, results were alike in two population-based studies with different aims, most of the participants were unaware of having a mild grade anemia [1], and mortality risk remained almost the same when two-thirds of non-participants of the H&A 65–84 were added to the previous short-term analyses [16]. Since prevalent cases have had the condition for some time before the study starts, analyzing prevalent cohorts, as almost exclusively done in the available literature, assumes that the individuals are allocated to the anemia group at the time of blood drawing, while only prevalent cases that are alive at that time are actually included in the analyses. These cases, being less susceptible to the exposure, represent the selected healthier sub-group surviving to the beginning of the study (selective survival). Not including the subset of prevalent cases at risk before baseline but that fails to survive until the sampling date leads to a study population biased toward favorable survival and the risk associated with mild anemia found in prevalent cases would thereby be underestimated. In accordance with this consideration, mortality rate over the first seven years of follow up was lower among prevalent cases of anemia than among incident cases. A result also in agreement with that of the Leiden 85 study [11]. Consistently, elderly participants with thalassemia trait have a lifelong coexistence with mild anemia and are not at risk of a shorter survival, thus tempering the difference between mild anemia and non-anemia with respect to mortality. Furthermore, the present together with two others [11,29] are the only studies that investigated the association between anemia and change in hemoglobin concentration with mortality also in non-anemic cohorts. A further potential limitation of studies assessing hemoglobin concentration on a single occasion, is that they cannot address within-individual changes in hemoglobin concentration over time that may potentially affect the results. However, the present study is one of only two attempts [11] to investigate also whether a change in anemia/non-anemia status would affect the risk of death. Generally, being anemic or non-anemic is a rather consistent status over time at old age in the general population: in the present study 84% of the study sample with two blood draws continued to be anemic or non-anemic on average over two years. However, since "false positives" (due to physiological fluctuations) and those successfully treated would decrease the mortality rate of the anemic group in which they were initially classified, whereas the "false negatives" (due to physiological fluctuations) and incident cases would increase the mortality rate of the non-anemic group in which they were initially classified, the actual risk associated with the anemia status would tend to be underestimated in subjects with a single blood sample. Consistently, in both young-old and old-old cohorts, subjects with two hemoglobin determinations whose anemia "resolved" (whether they had been successfully treated after or "false positive" cases at first sampling) did not show a significantly increased risk of mortality compared to those who were consistently non-anemic, a result almost identical to that of the Leiden 85 study [11] and also of a retrospective study in a selected sample of patients with chronic heart failure [41]. Whereas in those who became anemic at second sampling (whether they had been incident cases after or "false negatives" at first sampling) the risk was increased with respect to prevalent cases, again in agreement with the Leiden 85 study [11]. Considering fluctuations in concentrations over time, it should also be noted that when the upper limit of hemoglobin concentration for anemia and/or the lower limit for mild anemia adopted were higher (Table 2), the estimated risks for the association between anemia/mild anemia and mortality were very similar to those of the main analysis. Another possible limitation, common to all the literature on the subject, is that covariates potentially associated with death as well as anemia/n0n-anemia status were assessed only at cohort entry, thus failing to account for the subsequent change over time of the covariates. However, restricted to the participants in the Monzino 80-plus study, the association between anemia and mortality could be further adjusted also for time-varying

covariates and change in anemia status over time: the results confirmed those adjusted only for baseline covariates and showed a slightly to moderately increased risk with respect to elderly subjects with a single blood sample and covariate assessment at entry, in agreement with the observation that changes in anemia/non-anemia status over time would tend to underestimate the risk assessed at baseline.

Findings from present and previous population-based studies strongly call into question the view of mild anemia as an innocent bystander at older ages and indicate the need to tackle this blood disorder. Within the context of global population aging, the growing number of older persons with mild anemia will also have a relevant, ever-greater impact on healthcare requirements and costs [42–44]. Although individuals with resolved anemia did not show a significant increased risk of mortality, present studies were observational. While keeping in mind Paltiel and Clarfield's word of caution about the large and growing target constituted by elderly people for interested parties seeking new markets [45], improving current understanding of the pathophysiological mechanisms underlying unexplained and chronic disease anemias would provide a potential basis for therapeutic interventions [7,9,36,46,47]. Randomized controlled trials could aid in establishing whether restoring normal hemoglobin concentrations by treating the specific causes of mild anemia at older ages could safely revert or reduce the observed risks.

## Supporting information

**S1 Fig. Flow chart of the *Health and Anemia* (*H&A*) and *Monzino 80-plus* (*M80+*) studies.**
(DOCX)

**S1 Table. Risk of mortality in anemic and mild anemic compared to non-anemic participants in pooled *Health & Anemia* and *Monzino-80 plus* population-based studies.**
(DOCX)

**S2 Table. Risk of mortality in anemic and mild anemic compared to non-anemic oldest-old participants in two population-based studies.**
(DOCX)

**S3 Table. Risk of mortality in anemic and mild anemic compared with non-anemic oldest-old participants aged 85 years or older in two pooled population-based studies.**
(DOCX)

**S4 Table. Risk of mortality in anemic and mild anemic compared with non-anemic participants aged 65–79 years at blood sample from the *Health and Anemia* population-based study and participants aged 80 years or older at blood sample from two pooled population-based studies (*Health and Anemia 80+* and *Monzino 80-plus*).**
(DOCX)

**S5 Table. Risk of mortality in anemic and mild anemic compared with non-anemic participants aged 65 years or older at blood sample from the *Health and Anemia 65+* population-based study and participants aged 80 years or older at blood sample from the *Monzino 80-plus* population-based study.**
(DOCX)

**S6 Table. Risk of mortality over 15 and 11 years in anemic and mild anemic compared with non-anemic participants aged 80 years or older at blood sample from the *Monzino 80-plus* population based study.**
(DOCX)

**S7 Table. Risk of mortality in anemic and mild anemic compared with non-anemic partici-pants aged 65–84 years at blood sample from the *Health and Anemia* population-based study and participants aged 80 years or older at blood sample from two pooled popula-tion-based studies (*Health and Anemia 85+* and *Monzino 80-plus*).**
(DOCX)

**S1 Methods.**
(DOCX)

## Acknowledgments

The authors would like to thank the participants of the *Monzino 80-plus* and *Health and Ane-mia* studies who have made these investigations possible.

We thank Mary Ellen Lanczak for language editing.

## Author Contributions

**Conceptualization:** Emma Riva, Mauro Tettamanti, Ugo Lucca.

**Data curation:** Alessia A. Galbussera, Sara Mandelli, Mauro Tettamanti.

**Formal analysis:** Alessia A. Galbussera, Sara Mandelli, Mauro Tettamanti.

**Funding acquisition:** Matteo G. Della Porta, Ugo Lucca.

**Investigation:** Alessia A. Galbussera, Sara Mandelli, Stefano Rosso, Roberto Zanetti, Marianna Rossi, Adriano Giacomin, Paolo Detoma, Emma Riva, Mauro Tettamanti, Matteo G. Della Porta, Ugo Lucca.

**Methodology:** Emma Riva, Mauro Tettamanti, Ugo Lucca.

**Project administration:** Ugo Lucca.

**Software:** Mauro Tettamanti.

**Supervision:** Emma Riva, Mauro Tettamanti, Ugo Lucca.

**Visualization:** Alessia A. Galbussera, Ugo Lucca.

**Writing – original draft:** Ugo Lucca.

**Writing – review & editing:** Alessia A. Galbussera, Sara Mandelli, Stefano Rosso, Roberto Zanetti, Marianna Rossi, Adriano Giacomin, Paolo Detoma, Emma Riva, Mauro Tetta-manti, Matteo G. Della Porta.

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
