## [Decision Letter · Decision Letter 0]

25 Aug 2021

PONE-D-21-13069

Mild anemia and 15-year mortality in young-old and old-old: Results from two population-based cohort studies

PLOS ONE

Dear Dr. Lucca,

Thank you for submitting your manuscript to PLOS ONE. After careful consideration, we feel that it has merit but does not fully meet PLOS ONE’s publication criteria as it currently stands. Therefore, we invite you to submit a revised version of the manuscript that addresses the points raised during the review process.

We look forward to receiving your revised manuscript.

Kind regards,

Laurent Azoulay, PhD

Academic Editor

PLOS ONE

Journal Requirements:

Additional Editor Comments (if provided):

This is an interesting study, but certain aspects of the study need to be clarified. Please address the reviewer comments.

Reviewers' comments:

Reviewer's Responses to Questions

**Comments to the Author**

1. Is the manuscript technically sound, and do the data support the conclusions?

Reviewer #1: No

Reviewer #2: Yes

2. Has the statistical analysis been performed appropriately and rigorously? 

Reviewer #1: No

Reviewer #2: Yes

3. Have the authors made all data underlying the findings in their manuscript fully available?

Reviewer #1: No

Reviewer #2: No

4. Is the manuscript presented in an intelligible fashion and written in standard English?

Reviewer #1: Yes

Reviewer #2: Yes

5. Review Comments to the Author

Reviewer #1: This study follows two Italian cohorts to examine the association of mild anemia and all cause mortality. The topic is important, but here are some aspects to be considered.

1. “Objective of the present study was to prospectively investigate the long-term effect of mild anemia and mild anemia types on all-cause mortality in the young-old (65-84 years) and old-old (80+ years) from two population-based studies.”

This is a causal question. The methods in this paper do not allow a clear causal interpretation (and it has little to do with this being a cohort study). I’d suggest that the use of the term “effect” is reconsidered.

2. The setting up of the two cohorts is confusing. Seems that the distinguishing features of the final cohorts are cohort entry age. The 65-84 year olds comprise participants from H&A65-84, while 80+ comprise participants from both M80+ and H&A65-84. Some 80+ individuals, thus, are present in the 65-84 year olds cohort, and others in the 80+ cohort. What is the rationale? I feel that this muddles the interpretation of the effect estimates, obscuring the effect of both age and location. Why not, either have H&A and M80+ as the two distinct cohorts, or have 65-80 and 80+ as two distinct cohorts.

3. I think it might be useful to explicitly define cohort entry. To me it seems that participants entered the cohort upon providing blood samples, in 2003 or 2007 (H&A) or 2002 or2009-2019 (M80+). Is this correct? This of course results in the major problem of prevalent anemia, which is only partially accounted for in the incident anemia secondary analysis. This needs to be discussed as a major limitation.

4. How hemoglobin was measured should be mentioned in methods.

5. How were the various types of anemia classified?

6. When exactly were the questionnaires administered and informed consent obtained: at cohort entry, before, after? I’d have thought that the history of hematological disorders is a critical covariate in this analysis, which is missing

7. It is important to get a sense of the person time followed up, and the incidence rates of the events of interest, which have not been mentioned. What was the median follow-up? Were individuals who were not at risk at 8-end of follow-up excluded from the survival analysis from 8 to 15/11 years?

8. The assumption in the main analysis, where there is no censoring except for at event/administratively, is that once anemia is detected, it irreversibly affects the risk of death. How justified is this (particularly given that the authors conduct an analysis where they examined Anemia No Anemia patients)? If it is not entirely justified, then this should be mentioned as a limitation.

9. Another major assumption is that the covariates that predict death do not change between cohort entry and end of follow-up. This of course is not justified, and should be added to the limitations. On the issue of adjustment, the assertion that “Extensive adjustment may have led to underestimation of the association strength, since mild anemia could also be an effect of underlying pathological conditions,” is problematic given that the authors are interested in causal estimates. If after adjusting the estimates move towards the null, then that is the real effect. Had they been interested in mortality burden in a non-etiological sense, crude rates would have been important.

10. About 50% participated in each cohort. The authors report data on similarity in variable distribution between those who did and did not participate. However, to infer whether there was selection bias, one needs to know whether there were differences in variables that affect both anemia and mortality risk. As such, only differences in age, sex, and prevalence in dementia. But surely other variables may be associated with anemia and mortality risk that may be differentially distributed between participants and non-participants. For example, what about diabetes, htn, cancer, etc.

10. How was the stratified randomization for the second venous sampling conducted, on survivors or on all initial participants. How many of the initial participants died during this time?

11. The incidence rates and person time follow up should be mentioned for all analyses.

12. What are the reference group in the analyses reported in table 4?

13. It was surprising to see such a strong HR among No anemia Anemia. Again rates will help clarify this.

14. “In these two large prospective population-based studies, elderly persons with mild anemia had an overall 40% increased risk of dying”: mentioning a single estimate without meta-analyzing is confusing.

15. “The risk was somewhat higher in young-old (63%) than in old-old (38%), clearly due to a life expectancy twice as long at age 65 (18.5 years) than at age 80 years (8.5 years) (2003-5)”. Not very clear to me how this follows the life expectancy estimate. Do the authors mean that at an younger age there are fewer causes of death and that anemia is a greater contribution to mortality risk and so the risk is higher? If so, this is a good argument, but needs elaboration. Otherwise, please clarify.

16. KM in 80+ separates immediately after cohort entry, indicating confounding/prevalent exposure related bias. Otherwise kindly justify mild anemia may lead to death within 2 months of follow-up

17. “Cohort studies cannot determine causality”, “Only randomized controlled trials could finally establish whether restoring normal hemoglobin concentrations by treating the specific causes of mild anemia at older ages could safely revert or reduce the observed risks”: neither statement is true, consider changing.

Reviewer #2: Summary

This study examined whether mild anemia is associated with increased all-cause mortality in individuals aged 65-84 years and 80+ between 2002/2003 and 2017/2018. Two cohorts were constructed (H&A 65-84 and H&A 85+, M80+), with follow-up of 15, 11, and 15 years, respectively. During the study period, mild anemia was associated with a higher hazard of all-cause mortality, compared with those without anemia (HR: 1.42, 95% CI 1.22-1.65 in H&A 65-84 and HR: 1.32, 95% CI: 1.18-1.48 in H&A 85+ and M80+).

Abstract

- Please clarify the results section of the abstract as it provides results of the 0-7 years analysis although that analysis is not mentioned in the methods section of the abstract. Additionally, only the 0-7 years results are presented rather than the full results. Further, the title mentioning the 15-year mortality should be modified considering that one of the cohorts has a maximum follow-up of 11 years.

Introduction

- Please add more information on anemia i.e., is this a transient vs long-lasting “exposure”? If transient, is anemia thought to have lasting effects?

Study settings and participants

- Although the authors provide references to previous publications for details of study population and design, this manuscript should include relevant details. For example, what are the data collection time points to assess changes in anemia status? In the study design section, the manuscript only specifies that “venous blood samples were collected at the place of residence” and that a random sample from the H&A 65-84 study were contacted in 2005/2006 to collect blood samples and consenting individuals in the M80+ cohort were asked “at following visits” to collect blood samples. This section should clearly specify the number of collections that occurred for the cohorts as well as the time frame, rather than only presenting the information in the Results section.

Study design

- The section mentions that venous blood samples were collected at the place of residence. Please clarify that this exposure was collected at the beginning of the study and thus marks the start of follow-up. Importantly, the limitation section should discuss how capturing prevalent cases of mild anemia (vs. incident) might impact the results.

- The questionnaires were administered by trained registered nurses in the H&A cohort and by psychologists in the M80+ cohort. Would the questionnaire responses expected to be different in the two cohorts based on the different assessors? Please also clarify the sentence on the agreement between interviewers on medical history queries being very high – were there more than one interviewer per individual?

Statistical analysis

- It is unclear whether data from the H&A 85+ cohort is analyzed with the M80+. The previous sections do not mention the pooling of the older cohorts. It is only first mentioned in this section that the data for the H&A 85+ cohort and the M80+ cohort had been pooled. Please clarify specifically and earlier that pooling occurred and how, and discuss potential impacts of pooling these two cohorts given different data collection processes, different follow-up times etc.

- For the analysis of the effect of mild anemia over time, how was person-time handled in the analysis? Were individuals contributing to the 0-7 years group up until they reached past 7 years of follow-up?

- For the analysis re-defining the criteria for mild anemia, please indicate the year that the changes in lower limits and WHO criteria were proposed.

- Please indicate in the flow chart how many individuals had further blood collection.

- Were the individuals who refused blood collection at baseline different than those who did not, aside from age and sex (e.g., comorbidities)?

Results

- It appears that no individuals in the M80+ cohort were excluded due to death/not found as in the H&A cohort. If the 2,039 individuals were deemed eligible because they were alive at first interview, please add this detail to the first box of the flow chart.

- This section mentions that the primary research targets were dementia and cognitive function. This is the first time the reader comes across this information. If those were the primary aims of the original cohorts, please specify briefly under “Study settings and participants”.

- For this study, the analyses are based on the exposure of mild anemia vs non-anemia. Table 1 should then present the baseline characteristics of individuals for each of these exposure groups.

- If exposure was assessed only at baseline, it is possible that the study captured both prevalent and incident cases of anemia. Capturing prevalent cases can be problematic especially when assessing the association between an exposure and mortality. For example, some individuals might have had mild anemia for several years prior to the baseline data collection. As mentioned before, these limitations must be discussed.

- It is unclear what the time axis is for the cohort. In the text, it appears that it is duration of follow-up in years, although the KM curves show time to death. What were the censoring events?

- What was the covariate assessment window?

- What was the mean follow-up time?

- Was the type of anemia (i.e., anemia due to specific conditions) ascertained at baseline? If so, please indicate in the methods section. How can we be sure that the increased risk seen for anemia type is not due to the underlying condition, and what is the impact of adjusting for some of these comorbidities if they are also included in the exposure definition for anemia type?

- With several criteria used in this study to define mild anemia, please provide the rationale in the methods section for the choice of cut-off used to report the primary results.

- This section mentions pooling the oldest-old cohorts because the hazard ratios were similar. This seems like an ad-hoc decision, which should not have been based on the HRs. Please provide a clearer rationale as to why the two cohorts were pooled in the first place, considering that they have different follow-up times and data collection methods.

- A second sample was only available for 15.4% of subjects in the H&A 65-84 cohort. Were the individuals with a second blood sample different from those who did not have a second blood sample?

- For the analysis capturing the second blood samples, were patient characteristics and comorbidities measured at baseline or at the time of the second blood sample?

- It is worrisome that anemia was measured only at baseline (a time point not defined by any specific event), which included both prevalent and incident cases, and that the relationship between this exposure on mortality was assessed over up to 15 years later. Please discuss the impact of choosing an exposure definition that only captures the baseline status without accounting for changes in the exposure status or comorbidities over time, and includes prevalent and incident cases. Individuals included in the mild anemia group might not have been anemic for a large portion of the follow-up time, and vice versa.

Discussion

- The “dose-response relationships” are only stated in the abstract and discussion. This definition of the analysis should be explicitly stated in the methods (…to assess whether a dose-response relationship exist…”) and results. Further, it is unclear whether this dose relationship refers to the different cut-off criteria or the comparison of HRs between anemia and mild anemia (which, in the case of anemia, is based on very few individuals with non-mild anemia).

6. PLOS authors have the option to publish the peer review history of their article (what does this mean?). If published, this will include your full peer review and any attached files.

Reviewer #1: No

Reviewer #2: No

---

## [Author Response · Author response to Decision Letter 0]

19 Oct 2021

Editor Comments: "This is an interesting study, but certain aspects of the study need to be clarified. Pleas address the reviewer comments".

A point-by-point response to all of the Reviewers' comments together has been submitted (attached file).

Response to Reviewers

Reviewer #1: This study follows two Italian cohorts to examine the association of mild

anemia and all-cause mortality. The topic is important, but here are some aspects to be

considered.

We are grateful to the Reviewers for the several challenging issues they raised which

gave us the opportunity to improve the quality of our manuscript.

1. “Objective of the present study was to prospectively investigate the long-term effect

of mild anemia and mild anemia types on all-cause mortality in the young-old (65-84

years) and old-old (80+ years) from two population-based studies.”

This is a causal question. The methods in this paper do not allow a clear causal

interpretation (and it has little to do with this being a cohort study). I’d suggest that the

use of the term “effect” is reconsidered.

Response

Following the Reviewer’s suggestion we have changed the term “effect” with the more

appropriate “association”:

“Objective of the present study was to prospectively investigate the long-term

association of mild anemia and mild anemia types on all-cause mortality in the youngold

(65-84 years) and old-old (80+ years) from two population-based studies.”

2. The setting up of the two cohorts is confusing. Seems that the distinguishing features

of the final cohorts are cohort entry age. The 65-84 year olds comprise participants from

H&A65-84, while 80+ comprise participants from both M80+ and H&A65-84. Some

80+ individuals, thus, are present in the 65-84 year olds cohort, and others in the 80+

cohort. What is the rationale? I feel that this muddles the interpretation of the effect

estimates, obscuring the effect of both age and location. Why not, either have H&A and

M80+ as the two distinct cohorts, or have 65-80 and 80+ as two distinct cohorts.

Response

We aimed to study the long-term association between mild anemia and mortality in both

the young-old and the old-old. This latter age group is one on which very few data are

available or almost absent if mild anemia instead of anemia of any grade is considered.

The rationale behind our choice was consequently to investigate this association in two

sufficiently large cohorts of young-old and old-old from the general population. The

Monzino 80-plus (M80+) is a study specifically designed to study the oldest-old, even

though the primary objective was not to explore anemia/mild anemia. The Health and

Anemia (H&A) study was performed in two distinct stages well mirroring the above

categorization of old age: as reported under “Study settings and participants”, “All

registered individuals aged 65 to 84 years in 2003 were eligible (study years: 2003-

2018). In 2007 the study was extended to all residents 85 years or older (study years:

2007-2018)”. Moreover, data on the association of mild-anemia with mortality in the

young-old cohort (65-84 years) over the first 3.5 years have already been published

(reference 16). Therefore, we thought it was sounder not to modify the composition of

the original H&A 65-84 cohort by moving its 80-84 age group to the subsequent H&A

85+ cohort.

The old-old are variously defined as persons aged 80 or 85 years and older. In this

regard we performed several sensitivity analyses (Supplementary Tables S1, S2, and

S3) and all gave comparable results. We have now also analyzed the four cohorts the

Reviewer suggested (i.e.: H&A65-79, [H&A80+ and M80+], H&A65+, and M80+) and

the results are very similar to those reported in the original Table 2.

We have now also reported the results of the suggested cohort analyses in the

supplementary material.

Sentence added to sensitivity analyses:

“Results were akin also setting-up different cohorts: H&A65-79 (S4 Table), H&A80+

together with the M80+ (that is all individuals aged 80 years or older; S4 Table),

H&A65+ (S5 Table), or M80+ (S5 Table).”

3. I think it might be useful to explicitly define cohort entry. To me it seems that

participants entered the cohort upon providing blood samples, in 2003 or 2007 (H&A)

or 2002 or2009-2019 [2010?] (M80+). Is this correct? This of course results in the

major problem of prevalent anemia, which is only partially accounted for in the incident

anemia secondary analysis. This needs to be discussed as a major limitation.

Response

The Reviewer is correct: participants entered the cohorts analyzed in the present study

upon providing blood samples (even though for the Monzino 80-plus study the date of

the first main study visit preceded that of the blood draw [please see answer to comment

6]). We have now added your suggestion under the “Study design”:

“All participants entered the cohorts on the date of their blood draw”.

Mild anemia is a condition most elderly persons do not even realize that they have until

it is unexpectedly identified in a complete blood count (likely the most common among

the routine blood tests) and even in that eventuality it is often disregarded in everyday

clinical practice. These prevalent cases are exactly those seen in general practice. Even

though in our view prevalent anemia does not represent a major problem, the present

study has investigated the association between anemia and mortality also in a

prospective cohort of young-old and old-old without anemia. Investigating the

association of incident anemia with mortality was not a secondary analysis (“to assess

the effect of change in anemia status …”), rather one of the main and novel aims of the

present study (all the available literature investigated this association in prevalent cases

of anemia and only two among these studies also in incident cases, one in a selected

population of young old [29] and one in a cohort of 85-years-olds [11].

Prevalent cases have actually had anemia for a longer period of time than that calculated

from blood sampling. However, prevalent cases, being less susceptible to the exposure,

represents the selected healthier sub-group surviving to the beginning of the study

(selective survival) and the risk associated with mild anemia found in prevalent cases

would thereby be underestimated. In accordance with this explanation, mortality rate

over the first seven years of follow up was lower among prevalent cases of anemia than

among incident cases. A result in agreement with that of the Leiden 85 study (mortality

rates not reported): prevalent anemia (follow-up: 5 years): HR: 1.41 (1.13-1.76);

incident anemia (follow-up: ? years [< 5 years]): HR: 2.08 (1.60-2.07) [reference 11]).

Consistently, elderly subjects with thalassemia trait have a lifelong coexistence with

mild anemia and are not at risk of a shorter survival, thus tempering the difference

between mild anemia and non-anemia with respect to mortality. In fact, if the 63

participants with thalassemia trait were removed from the analysis of the H&A65-84

cohort (the 7 thalassemic subjects in the H&A 85+ are too few for any further analysis),

all HRs over 0-15, 0-7 or 8-15 years of follow-up would consistently increase with

respect to those analyzing the complete cohort.

For the problem of incorrect classification or changing anemia status over time when

hemoglobin is determined only once as in prevalent cases, please also see the answer to

comment 8.

Following the Reviewer’s request we have discussed the point raised under the

limitation part of the Discussion:

“Since prevalent cases have had the condition for some time before the study starts,

analyzing prevalent cohorts, as almost exclusively done in the available literature,

assumes that the individuals are allocated to the anemia group at the time of blood

drawing, while only prevalent cases that are alive at that time are actually included in

the analyses. These cases, being less susceptible to the exposure, represent the selected

healthier sub-group surviving to the beginning of the study (selective survival). Not

including the subset of prevalent cases at risk before baseline but that fails to survive

until the sampling date leads to a study population biased toward favorable survival and

the risk associated with mild anemia found in prevalent cases would thereby be

underestimated. In accordance with this consideration, mortality rate over the first seven

years of follow up was lower among prevalent cases of anemia than among incident

cases. A result also in agreement with that of the Leiden 85 study [11]. Consistently,

elderly participants with thalassemia trait have a lifelong coexistence with mild anemia

and are not at risk of a shorter survival, thus tempering the difference between mild

anemia and non-anemia with respect to mortality. Furthermore, the present together

with two others [11,29] are the only studies that investigated the association between

anemia and change in hemoglobin concentration with mortality also in non-anemic

cohorts”.

Under sensitivity analysis:

“If the 63 participants with thalassemia trait (62 with mild and 1 with moderate

anemia) were removed from the analysis of the H&A65-84 cohort, hazards of death

from mild anemia would consistently increase with respect to those analyzing the

complete cohort: fully-adjusted HRs (95% CIs): 1.45 (1.22-1.72) over 0-15 years, 1.78

(1.41-2.25) over 0-7 years, and 1.21 (0.94-1.57) over 8-15 years”.

4. How hemoglobin was measured should be mentioned in methods.

Response

We have now added the methods used to measure hemoglobin at the beginning of the

“Definitions of anemia and mild anemia” section:

“Hemoglobin concentration together with the other tests included in the complete blood

count were determined on automated hematology analyzer instruments at the laboratory

of Biella Hospital (H&A) and Laboratorio Milano, Milano (M80+)”.

5. How were the various types of anemia classified?

Response

As indicated at the end of the “Definitions of anemia and mild anemia” section,

definitions used to classify the anemia types were reported under the Supplementary

methods of the Supporting Information.

6. When exactly were the questionnaires administered and informed consent obtained:

at cohort entry, before, after? I’d have thought that the history of hematological

disorders is a critical covariate in this analysis, which is missing

Response

Informed consents were signed just before blood sampling in both studies.

Questionnaires were administered after blood sampling in the H&A study. In the M80+

study, blood sampling was carried out in an ad hoc additional visit at the place of

residence on average within 1.9 months of the scheduled study visit during which, after

the administration of the questionnaire and tests, the subject's willingness to participate

in the blood sub-study (which was not the primary objective of the study for which

another specific informed consent was previously signed) was investigated.

Changes made to the original manuscript:

At the end of “Study setting and participants”: “Written informed consent was obtained

from participants before blood sampling. Participants in the M80+ study had also

previously signed the informed consent to participate in the main study. Written

informed consent was also obtained from informants in both studies”.

Under “Study design”: “Questionnaires were administered after blood sampling in the

H&A study. In the M80+ study, blood sampling was carried out in an ad hoc additional

visit at the place of residence on average within 1.9 months of the scheduled study visit

during which, after the administration of the questionnaire and tests, the subject's

willingness to participate in the blood sub-study was investigated. The questionnaire

was administered by …”

Prevalence of hematological diseases other than anemia is low (about 2% in the present

study) and not necessarily associated with mortality. Moreover, in the present study,

about 65% of the hematological disorders other than anemia are represented by

malignant neoplasms of the lymphoid and hematopoietic tissues which were already

classified under “history of cancer”.

7. It is important to get a sense of the person time followed up, and the incidence rates

of the events of interest, which have not been mentioned. What was the median followup?

Were individuals who were not at risk at 8-end of follow-up excluded from the

survival analysis from 8 to 15/11 years?

Response

Yes, individuals who were not at risk (i.e. those who died) by the end of the first period

considered (0-7 years) were excluded from the analysis of the second period (8-15 or 8-

11 years).

We have now specified under “Statistical analysis” that

“To examine whether the effect of mild anemia was similar over time, two survival

analyses were set up: from 0 to 7 years, and, in participants who survived the first seven

years of follow up, from 8 to 15/11 years”.

We have also added the requested information on “person time followed up, and the

incidence rates of the events of interest”. Please note that, in order to make the inserted

parts intelligible, we report the whole beginning of the edited part:

“H&A65-84 cohort

During 15 years of follow-up after blood sampling, 230 anemic (66.9%; 205 mild

anemic) and 1928 non-anemic (46.5%) individuals died (Fig 1). The median follow-up

period was 14.0 years with 50,522 person-years of observation and a mortality rate of

4.3 per 100 person-years (4.1 in non-anemic and 7.1 in mild anemic participants). Over

the 15-year follow-up, mortality risk was significantly increased in participants with

anemia (fully-adjusted HR: 1.42; 95% CI, 1.22-1.65) and mild anemia (fully-adjusted

HR: 1.35; 95% CI, 1.15-1.58). In the first seven years after blood collection (median

follow-up: 7.0 years; person-years of observation: 28,607 years; incidence mortality

rate: 2.9 per 100 person-years), compared with non-anemic … . From 8 to 15 years

(3,574 participants; median follow-up: 7.4 years; person-years of observation: 21,915;

incidence mortality rate: 6.1 per 100 person-years) the risk was still … .

…

H&A85+ and M80+ cohorts

...

During 11 years of follow-up, 526 anemic (94.1%; 449 mild anemic) and 1,137 nonanemic

individuals (88.6%) died in the pooled 80+ cohort (Fig 1). The median followup

period was 3.5 years with 7,871 person-years of observation and a mortality rate of

21.1 per 100 person-years (18.7 in non-anemic and 28.6 in mild anemic participants).

Over the 11-year follow-up, mortality risk was significantly higher in participants with

anemia (fully-adjusted HR: 1.32; 95% CI, 1.18-1.48) and mild anemia (fully-adjusted

HR: 1.28; 95% CI, 1.14-1.44). In the first seven years after blood collection (median

follow-up: 3.5 years; person-years of observation: 7,235 years; incidence mortality rate:

19.7 per 100 person-years), compared with non-anemic, … . From 8 to 15 years (287

participants; median follow-up: 2.3 years; person-years of observation: 637; mortality

rate: 23.9 per 100 person-years) no significant difference … .

…

Risk of mortality associated with mild anemia in “healthy” elderly subjects

At baseline 1093 elderly persons (mean age 74.6, 64% women) from both H&A65+ and

M80+ had no history of any of the diseases entered as confounders in multivariable

analyses. The median follow-up period was 7 years with 7,043 person-years of

observation and a mortality rate of 2.5 per 100 person-years (2.2 in non-anemic and 7.6

in mild anemic participants). Individuals with mild anemia (N = 74) showed an

increased risk of mortality … .

Risk of mortality associated with anemia status assessed over time (under “Study

design” or “Results”):

In H&A65-84+, a second blood sample was available for 692 subjects … . The mean

(SD) time between samplings was 2.2 (0.1) years (median follow-up: 7.0 years; personyears

of observation: 4,265; mortality rate: 3.8 per 100 person-years). …

Considering the 523 subjects without anemia at first sampling in H&A65-84+ (median

follow-up: 7.0 years; person-years of observation: 3,286; incidence mortality rate: 3.3

per 100 person-years), the 27 incident cases (5.2%; mortality rate: 11.4 per 100 personyears)

… . … .

In M80+, a second blood sample was available for 366 subjects … . The mean (SD)

time between samplings was 1.7 (1.0) years (median follow-up: 3.1 years; person-years

of observation: 1,298; incidence mortality rate: 22.4 per 100 person-years). …

Considering the 279 subjects without anemia at first sampling in M80+ (median followup:

3.4 years; person-years of observation: 1,048; mortality rate: 20.5 per 100 personyears),

the 65 incident cases at second sampling (23.3%; mortality rate: 31.8 per 100

person-years) showed an increased risk of mortality during the following 7 years

compared to those who did not develop anemia (fully-adjusted HR: 1.47; 95% CI, 1.04-

2.07). …

8. The assumption in the main analysis, where there is no censoring except for at

event/administratively, is that once anemia is detected, it irreversibly affects the risk of

death. How justified is this (particularly given that the authors conduct an analysis

where they examined Anemia®�No Anemia patients)? If it is not entirely justified, then

this should be mentioned as a limitation.

Response

“In detecting and evaluating an anemia problem in a community, reference standards

are necessary, even though they may be somewhat arbitrary” (WHO 1968). Several

factors can influence the level of hemoglobin measured (for example, physiological

fluctuations of plasma volume), especially in population-based studies. Since the

diagnosis of anemia is defined as a concentration of hemoglobin of less than a

conventional lower limit of normal, physiological fluctuations of hemoglobin

concentrations can be incorrectly classified as either “anemia” (“false positive”) or

“non-anemia” (“false negative”) when hemoglobin is determined only once. Moreover,

among those diagnosed with an anemia type susceptible to treatment, a fraction can

return over time to having a normal concentration of hemoglobin when successfully

treated or the underlying causes have been eliminated. Whatever the assumption,

considering how anemia is defined and the existence of anemia types susceptible of

treatment, imply that anemia is not necessarily a chronic, everlasting condition.

Moreover, please note that the present study has investigated the association between

anemia and mortality also in prospective cohorts of young-old and old-old with two

samplings and reported the risk associated precisely with change in anemia status

(Table 4).

Please, see also the answer to comment 3.

Following the Reviewer’s request we have now added this point to the study

limitations:

“A further potential limitation of studies assessing hemoglobin concentration on a

single occasion, is that they cannot address within-individual changes in hemoglobin

concentration over time that may potentially affect the results. However, the present

study is one of only two attempts [11] to investigate also whether a change in

anemia/non-anemia status would affect the risk of death. Generally, being anemic or

non-anemic is a rather consistent status over time at old age in the general population:

in the present study 84% of the study sample with two blood draws continued to be

anemic or non-anemic on average over two years. However, since “false positives” (due

to physiological fluctuations) and those successfully treated would decrease the

mortality rate of the anemic group in which they were initially classified, whereas the

“false negatives” (due to physiological fluctuations) and incident cases would increase

the mortality rate of the non-anemic group in which they were initially classified, the

actual risk associated with the anemia status would tend to be underestimated in

subjects with a single blood sample. Consistently, in both young-old and old-old

cohorts, subjects with two hemoglobin determinations whose anemia “resolved”

(whether they had been successfully treated after or “false positive” cases at first

sampling) did not show a significantly increased risk of mortality compared to those

who were consistently non-anemic, a result almost identical to that of the Leiden 85

study [11] and also of a retrospective study in a selected sample of patients with chronic

heart failure (risk ratio 0.98 [0.73-1.36] [Tang et al. J Am Coll Cardiol 2008;51:569-

576][41]. Whereas in those who became anemic at second sampling (whether they had

been incident cases after or “false negatives” at first sampling) the risk was increased

with respect to prevalent cases, again in agreement with the Leiden 85 study [11].

Considering fluctuations in concentrations over time, it should also be noted that when

the upper limit of hemoglobin concentration for anemia and/or the lower limit for mild

anemia adopted were higher (Table 2), the estimated risks for the association between

anemia/mild anemia and mortality were very similar to those of the main analysis”.

Under the Results and in Table 4 we have now added also mortality risks associated

with hemoglobin decline (per 1 g/dL decline) in the young-old and old-old with no

anemia at baseline:

“In this cohort of young-old without anemia, also hemoglobin decline over the same

period after the second sampling was associated with an increased risk of mortality:

fully-adjusted (also for hemoglobin at first sampling) HR: 1.39 (95%CI, 1.10-1.76) per

1 g/dL decrease of hemoglobin concentration (Table 4). … In this cohort of old-old

without anemia, also hemoglobin decline over the same period after the second

sampling was associated with an increased risk of mortality (Table 4)”.

Under “Discussion”:

“…incident anemia and decline in hemoglobin concentration were significantly

associated with ….”

Under “Abstract-Results”:

“In participants without anemia at baseline hemoglobin decline was also significantly

associated with an increased mortality risk over seven years in both young-old and oldold”.

9. Another major assumption is that the covariates that predict death do not change

between cohort entry and end of follow-up. This of course is not justified, and should be

added to the limitations. On the issue of adjustment, the assertion that “Extensive

adjustment may have led to underestimation of the association strength, since mild

anemia could also be an effect of underlying pathological conditions,” is problematic

given that the authors are interested in causal estimates. If after adjusting the estimates

move towards the null, then that is the real effect. Had they been interested in mortality

burden in a non-etiological sense, crude rates would have been important.

Response

Since the comorbidities considered as covariates are all chronic diseases or have chronic

complications (e.g. stroke and myocardial infarction), the assumption is limited to the

new occurrence (incidence) of these diseases after baseline. Please also note that this

assumption as well is common to all the literature on the subject (and, incidentally, none

of the published studies has mentioned it among the limitations). However, we agree

with the Reviewer.

In the Monzino 80-plus study, beyond baseline, another 8 follow-up assessments were

available. Thus, limited to this study, it has been possible to further investigate the

influence of time-varying covariates on the association between anemia and mortality in

subjects with one (prevalent cases) as well as in subjects with two hemoglobin

determinations (incident cases and hemoglobin decline). Moreover, in subjects with two

hemoglobin determinations it was also possible to investigate how the association

between baseline anemia and mortality was affected by change in anemia status (at

second sampling) together with prevalent plus incident covariates over the following

seven years of follow-up. We have now reported the results of these analyses in a new

Table 5 (please, see the revised manuscript): all results confirmed those adjusted only

for baseline covariates previously set out in Tables 2 and 4 and in the new

Supplementary Table reporting only the results of the Monzino 80-plus study. Though

slightly, all hazard ratios are consistently higher than those reported in the above

mentioned Tables and Supplementary Table. Moreover, in the analysis where it has

been possible to account for change in both anemia status and covariates, the risk

associated with baseline anemia resulted moderately increased also with respect to the

model further adjusted for change in covariates. This finding is in agreement with the

observation that changes in anemia/non-anemia status over time would tend to

underestimate the risk of mortality associated with anemia in subjects with a single

blood sample (please see the answer to the previous comment).

Following the Reviewer’s request we have now added this point to the study

limitations:

“Another possible limitation, common to all the literature on the subject, is that

covariates potentially associated with death were assessed only at cohort entry, thus

failing to account for the subsequent change over time of the covariates. However,

restricted to the participants in the Monzino 80-plus study, the association between

anemia and mortality could be further adjusted also for time-varying covariates and

change in anemia status over time and the results confirmed those adjusted only for

baseline covariates showing a slightly to moderately increased risk with respect to

elderly subjects with a single blood sample and covariate assessment at entry, in

agreement with the observation that changes in anemia/non-anemia status over time

would tend to underestimate the risk assessed at baseline.”

Under “Statistical analysis” we have added:

“All covariates were assessed at baseline. Limited to the Monzino 80-plus study,

beyond baseline, another 8 follow-up assessments were available. It was thus possible

to further adjust the multivariable model also for the influence of time-varying

covariates (age, habits, and intervening comorbidities) on the association between

anemia and mortality in subjects with one (prevalent cases) as well as two hemoglobin

determinations (incident cases and hemoglobin decline). Moreover, in subjects with two

hemoglobin determinations it was also possible to investigate how the association

between baseline anemia and mortality was affected by time-varying covariates together

with anemia/non-anemia status at second sampling”.

Under “Results”, “H&A85+ and M80+ cohorts”:

“ … was found (Table 2). Limited to the M80+ cohort, mortality risk associated with

prevalent anemia and mild anemia was slightly increased when the model was further

adjusted for time-varying covariates (anemia: HR 1.59 [95% CI: 1.39-1.82] over 11

years and HR 1.64 [95% CI: 1.42-1.89] over seven years; mild anemia: HR1.57 [95%

CI: 1.36-1.80] over 11 years and HR 1.60 [95% CI: 1.38-1.85] over seven years)”.

Under “Results”, “Risk of mortality associated with anemia status assessed over time”,

at the end of the last paragraph:

“In all subjects with two hemoglobin determinations it was possible to control for both

change in the anemia/non-anemia status together with time-varying covariates: the risk

associated with baseline anemia resulted moderately increased also with respect to the

model further adjusted only for time-varying covariates (Table 5). In non-anemic

subjects with two hemoglobin determinations, the risk of mortality associated with

incident anemia or change in hemoglobin concentration over the seven years following

the second sampling was increased when also time-varying covariates were added to the

“fully”-adjusted model (Table 5).”.

Under “Discussion”:

“In the Monzino 80-plus study the risk associated with prevalent and incident anemia

was even higher when the multivariable model could be further adjusted also for timevarying

covariates and, limited to all old-old with two blood samplings, for change in

anemia status at second sampling.”.

Under “Discussion”, strength:

“No previous study has attempted to adjust also for the influence of change in

anemia/non-anemia status and time-varying covariates”.

With regard to the discussion on the issue of the “extensive adjustment”, following the

Reviewer’s observation we have removed the statement in the revised manuscript.

10. About 50% participated in each cohort. The authors report data on similarity in

variable distribution between those who did and did not participate. However, to infer

whether there was selection bias, one needs to know whether there were differences in

variables that affect both anemia and mortality risk. As such, only differences in age,

sex, and prevalence in dementia. But surely other variables may be associated with

anemia and mortality risk that may be differentially distributed between participants and

non-participants. For example, what about diabetes, htn, cancer, etc.

Response

Probably there must be a misunderstanding. We compared participants and nonparticipants

only for age and sex simply because age and sex were/are the only two

variables available in the municipality registry offices also for non-participants (i.e.,

eligible residents who refused or were untraceable). At any rate, age and sex are two of

the main determinants of anemia and mortality and, at least for age, also of most

chronic diseases.

Having a different study aim, the eligible for the blood sub-study of the M80+ were not

all the eligible residents, rather the sub-population of those among all residents who

accepted to participate in the main study (some 90% of all the eligible residents in any

case). We also explored the distribution of dementia and cognitive function in

participants and non-participants in the blood sub-study not because these variables are

associated with mortality (a matter of fact) or anemia (under scrutiny; in the Monzino

study for example anemia was not significantly associated with an increased risk of

developing dementia [Lucca et al. Alzheimers Dement 2020; 16 (S10): abstract]), but, as

reported in the manuscript, because these were the primary research targets (in other

words, the traits under investigation), that is, those characteristics that could have

influenced the elderly subjects in the decision to take part or not in the study/sub-study

and thus possibly determine a self-selection bias. Since the entire cohort of old-old was

the result of the pooling of two cohorts (H&A85+ and M80+), the only available

information for the H&A85+ cohort and consequently for the entire 80+ cohort was that

regarding age and sex.

By the way, at initial visit in the M80+ study diabetes, hypertension, and cancer were

more prevalent among participants with blood sampling than among those without, even

though the difference was not statistically significant for diabetes (p = 0.203) and cancer

(p = 0.239).

10[bis]. How was the stratified randomization for the second venous sampling

conducted, on survivors or on all initial participants. How many of the initial

participants died during this time?

Response

Under “Study settings and participants” section, we have now specified the process of

stratified randomization:

“… recontacted during 2005-2006: initial participants were stratified at baseline in

anemic and non-anemic strata, all eligible consenting anemic and a random sample of

eligible consenting non-anemic participants were included in the study on the

association of mild anemia with cognitive, functional, mood, and QoL outcomes [21]”.

“How many of the initial participants died during this time?”

We have now reported under the “Study settings and participants” section (“Risk of

mortality associated with anemia status assessed over time”):

“In H&A65-84, a second blood sample …: … among the 344 consenting participants

with baseline anemia, 29 died and 146 were not found or withdrew their consent to

participate at the time of the second blood draw; among the 655 consenting participants

without anemia at baseline, 20 died and 112 were non traceable or withdrew their

consent to participate at the time of the second blood draw”.

For the M80+:

“Of the initial 1,115 participants, 366 accepted to donate a second blood sample (mean

age: 89.7; men: 23.0%; with anemia: 23.8%; mean Hb: 13.1 g/dL); 667 did not or could

not donate a second blood sample (mean age: 90.4; men: 26.5%; with anemia: 36.0%;

mean Hb: 12.7 g/dL) and 82 died before the next visit (mean age: 92; men: 28.1%; with

anemia: 57.3%; mean Hb: 11.8 g/dL). Of the 667 oldest-old with only one blood

sample, 620 continued to participate in the Monzino study but refused a second blood

sample; 30 could not be traced and 17 refused to continue to participate in the Monzino

study)”.

11. The incidence rates and person time follow up should be mentioned for all analyses.

Response

Please, see answers to comment 7 and related changes.

12. What are the reference group in the analyses reported in table 4?

Response

We have now slightly changed footnote “c” of Table 4:

“Reference group: subjects consistently non-anemic at both samplings” (instead of:

“with respect to subjects consistently non-anemic at both samplings”).

In the text it was already reported that “Considering the 523 subjects without anemia at

first sampling, the 27 incident cases … compared to those who did not develop anemia

… . When subjects were categorized according to anemia status at both samplings

(Table 4), compared to subjects constantly non-anemic (n = 496), …” (under

“Results”).

13. It was surprising to see such a strong HR among No anemia®�Anemia. Again rates

will help clarify this.

Response

We are not sure we have understood why “it was surprising to see such a strong [?] HR

among “No anemia®�Anemia”. If the surprise refers to the evidence of a stronger

association in incident cases than in prevalent ones, please see answers to comments 3

and 8. As noted above, our results are in agreement with those of the Leiden 85 study:

“We found that incident anemia in participants beyond the age of 85 years had an even

stronger impact on mortality than prevalent anemia at age 85.” (reference 11, p. 156).

Also Ishani et al. analyzing a selected sample of patients with heart failure participating

in the SOLVD trial found that prevalent anemia was associated with a 44% increase in

the hazard of all-cause mortality, whereas incident anemia with a 108% increase (J Am

Coll Cardiol 2005;45:391-399).

14. “In these two large prospective population-based studies, elderly persons with mild

anemia had an overall 40% increased risk of dying”: mentioning a single estimate

without meta-analyzing is confusing.

Response

The results of the pooled analyses of the H&A and Monzino studies were reported in

supplementary S2 Table (now S1), as pointed out in the “Results” section (line 257 of

the original manuscript): “S2 Table summarizes the results in the pooled young-old and

old-old cohorts”. We have now reported the main results of this pooled analysis also in

the manuscript:

“S2 Table summarizes the results in the pooled young-old and old-old cohorts.

Compared with non-anemic, mortality risk was significantly higher in elderly

individuals with mild anemia in the first seven years after blood collection (fullyadjusted

HR: 1.40; 95% CI, 1.26-1.57) as well as over the entire 11-year follow-up

period (fully-adjusted HR: 1.29; 95% CI, 1.17-1.43). From 8 to 11 years no significant

difference in mortality risk between anemic and non-anemic elderly persons was

found”.

15. “The risk was somewhat higher in young-old (63%) than in old-old (38%), clearly

due to a life expectancy twice as long at age 65 (18.5 years) than at age 80 years (8.5

years) (2003-5)”. Not very clear to me how this follows the life expectancy estimate. Do

the authors mean that at an younger age there are fewer causes of death and that anemia

is a greater contribution to mortality risk and so the risk is higher? If so, this is a good

argument, but needs elaboration. Otherwise, please clarify.

Response

We agree with the Reviewer’s criticism of our interpretation and have decided to

remove this sentence from the revised manuscript. Changes in the first sentence of the

“Discussion” section: “… an overall 40% increased risk of dying, 60% in young-old

and 38% in old-old”.

16. KM in 80+ separates immediately after cohort entry, indicating

confounding/prevalent exposure related bias. Otherwise kindly justify mild anemia may

lead to death within 2 months of follow-up

Response

Since not only mild anemia but multiple factors influence survival, by visualizing a

Kaplan Meir curve it is not possible to estimate the effect of a factor. Being a cohort of

very old, of course they start to die from the very beginning of the observed period, be

they subjects with (mean age 91.2 years) or without (mean age 89.4 years) anemia: after

one month 12 mild anemic and 14 non-anemic died, 21 and 22 respectively after two

months, 30 and 32 after three months, and 46 and 44 after fourth months, when the

difference between groups reached statistical significance at multivariable analysis.

Therefore, our interpretation is that the rapid separation is mainly due to the strong

background mortality, condition in which a hazard ratio can have a visible effect. With

regard to “confounding” bias, it should also be noted that Cox proportional hazard

models were adjusted for age, sex, education, smoking status, alcohol consumption,

hypertension, diabetes, heart failure, myocardial infarction, chronic respiratory failure,

chronic renal insufficiency, cancer, transient ischemic attack, stroke, parkinsonism,

dementia, hospitalization during the previous year, and study. And with regard to

“prevalent exposure related bias”, “KM in 80+ separate immediately after cohort entry”

also when incident cases versus consistently non-anemic cases were analyzed (see

Figure below). With regard to “confounding/prevalent exposure related bias”, please see

also the answers to comments 3, 8, and 9, and Table 5.

17. “Cohort studies cannot determine causality”, “Only randomized controlled trials

could finally establish whether restoring normal hemoglobin concentrations by treating

the specific causes of mild anemia at older ages could safely revert or reduce the

observed risks”: neither statement is true, consider changing.

Response

In compliance with the Reviewer’s observation, we have changed the two statements:

Under limitations:

“Although prospective cohort studies can help to assess a causal association, further

experimental trials would contribute to establish true causality.”

Last sentence of the conclusions:

“Randomized controlled trials could aid in establishing whether restoring normal

hemoglobin concentrations by treating the specific causes of mild anemia at older ages

could safely revert or reduce the observed risks”.

Reviewer #2: Summary

This study examined whether mild anemia is associated with increased all-cause

mortality in individuals aged 65-84 years and 80+ between 2002/2003 and 2017/2018.

Two cohorts were constructed (H&A 65-84 and H&A 85+, M80+), with follow-up of

15, 11, and 15 years, respectively. During the study period, mild anemia was associated

with a higher hazard of all-cause mortality, compared with those without anemia (HR:

1.42, 95% CI 1.22-1.65 in H&A 65-84 and HR: 1.32, 95% CI: 1.18-1.48 in H&A 85+

and M80+).

We are grateful to the Reviewers for the several challenging issues they raised which

gave us the opportunity to improve the quality of our manuscript.

Abstract

- Please clarify the results section of the abstract as it provides results of the 0-7 years

analysis although that analysis is not mentioned in the methods section of the abstract.

Additionally, only the 0-7 years results are presented rather than the full results. Further,

the title mentioning the 15-year mortality should be modified considering that one of

the cohorts has a maximum follow-up of 11 years.

Response

According to the guidelines the abstract should not exceed 300 words. Since mild

anemia, in both the young-old and the old-old, was not significantly associated with

mortality over the second part of the time period investigated (8-15 and 8-11 years), we

decided to report more conservatively the results of the first (0-7 years) instead of the

entire time period considered.

Following the Reviewer’s remarks, we have now changed the abstract results:

“…, mortality risk over 15/11 years was significantly higher in individuals with mild

anemia compared with those without (young-old: fully-adjusted HR: 1.35, 95%CI, 1.15-

1.58; old-old: fully-adjusted HR: 1.28, 95% CI, 1.14-1.44)”.

With regard to the title, we reported for conciseness only the 15-year mortality because

for the old-old there was also available a (comparable) finding over a 15-year time

period for the 895 participants aged 85+ years in the Monzino 80-plus study (see S1

Table: fully-adjusted HR: 1.38, 95% CI, 1.18-1.63) and now also for the entire Monzino

cohort of 1.115 subjects aged 80+ years (fully-adjusted HR: 1.36, 95% CI, 1.18-1.57).

In any case, in the Abstract it was already specified that “Objective of the study was to

investigate the association of mild anemia (…) with all-cause mortality over 11-15

years”.

Following the Reviewer’s remarks, we have also changed the title:

“Mild anemia and 11- to 15-year mortality in old-old and young-old: Results from two

population-based cohort studies”.

Introduction

- Please add more information on anemia i.e., is this a transient vs long-lasting

“exposure”? If transient, is anemia thought to have lasting effects?

Response

Is this a transient vs long-lasting “exposure”? Naturally anemia can be either a

temporary (for example, due to a bleeding, peri-operative blood loss, chemotherapy, in

general, any anemia that can be successfully treated or the underlying cause of which

can be eliminated) or a chronic disorder. The chronic is the most common condition

among the elderly population. In fact, “the management of anemias in older individuals

is a clinical challenge, especially when the etiology remains uncertain [one fourth to one

third of anemias in the elderly remains unexplained] and/or (multiple) comorbidities are

present” (Stauder et al. 2018). Besides, “in detecting and evaluating an anemia problem

in a community, reference standards are necessary, even though they may be somewhat

arbitrary” (WHO 1968) and several factors can influence the level of hemoglobin

measured (for example, physiologic fluctuations of plasma volume). Just because

anemia can be a temporary condition, the present study has investigated the association

between anemia and mortality also in a prospective cohort of young-old and old-old

with two samplings.

“If transient, is anemia thought to have lasting effects?” As far as we know, the

present study is the only one available investigating the association of mortality with

change in anemia status in the general population of both young-old and old-old. Except

for a study in a cohort of 85-year-olds (the Leiden 85 study, reference 11), we do not

know of any other population-based study investigating whether a temporary (mild)

anemia might have a lasting detrimental effect on health: the answer probably depends

on how long that temporary lasts and on the cause underlying the mild anemia (for

example, a congenital condition such as thalassemia trait would not seem to affect

survival at all).

For the unexpressed implication of the question, please see the answer to the first

comment under “Study design”…

Under introduction we have now added:

“Almost all studies assessed the relationship between anemia and mortality

exclusively in prevalent cases, who, however, were already exposed to the condition at

the time of blood sampling. Moreover, a single measurement of hemoglobin

concentration cannot investigate the association of change in anemia/non-anemia status

to subsequent mortality”.

Study settings and participants

-Although the authors provide references to previous publications for details of study

population and design, this manuscript should include relevant details. For example,

what are the data collection time points to assess changes in anemia status? In the study

design section, the manuscript only specifies that “venous blood samples were collected

at the place of residence” and that a random sample from the H&A 65-84 study were

contacted in 2005/2006 to collect blood samples and consenting individuals in the

M80+ cohort were asked “at following visits” to collect blood samples. This section

should clearly specify the number of collections that occurred for the cohorts as well as

the time frame, rather than only presenting the information in the Results section.

Response

Following the Reviewer’s requests we have specified the number of collections that

occurred for the cohorts as well as the time frame by moving the information from the

Result section to the “Study setting and participants” section:

“To assess change in anemia status, a stratified random sample of individuals from

H&A65-84 was recontacted during 2005-2006 [1]. In H&A65-84, a second blood

sample was available for 692 subjects (baseline: mean age 73.2 years, 55.4% women,

mean [SD] hemoglobin concentration 13.6 [1.7] g/dL, 24.4% anemic). The mean (SD)

time between samplings was 2.2 (0.1) years. … In M80+, participants who had

consented to donate a blood sample were asked at following visits whether they would

agree to a further blood sampling [19]. In M80+, a second blood sample was available

for 366 subjects (baseline: mean age 89.7 years, 77.1% women, mean [SD] hemoglobin

concentration 13.1 [1.5] g/dL, 23.8% anemic). The mean (SD) time between samplings

was 1.7 (1.0) years.”

Study design

- The section mentions that venous blood samples were collected at the place of

residence. Please clarify that this exposure was collected at the beginning of the study

and thus marks the start of follow-up. Importantly, the limitation section should discuss

how capturing prevalent cases of mild anemia (vs. incident) might impact the results.

Response

We have now added your suggestion under the “Study design” (second line):

“All participants entered the cohorts on the date of their blood draw”.

The Reviewer is correct: participants entered the cohorts analyzed in the present study

upon providing blood samples (even though for the Monzino 80-plus study the date of

the first main study visit preceded that of the blood draw [please see answer to comment

6]). We have now added your suggestion under the “Study design”:

“All participants entered the cohorts on the date of their blood draw”.

Mild anemia is a condition most elderly persons do not even realize that they have until

it is unexpectedly identified in a complete blood count (likely the most common among

the routine blood tests) and even in that eventuality it is often disregarded in everyday

clinical practice. These prevalent cases are exactly those seen in general practice. Even

though in our view prevalent anemia does not represent a major problem, the present

study has investigated the association between anemia and mortality also in a

prospective cohort of young-old and old-old without anemia. Investigating the

association of incident anemia with mortality was not a secondary analysis (“to assess

the effect of change in anemia status …”), rather one of the main and novel aims of the

present study (all the available literature investigated this association in prevalent cases

of anemia and only two among these studies also in incident cases, one in a selected

population of young old [29] and one in a cohort of 85-years-olds [11].

Prevalent cases have actually had anemia for a longer period of time than that calculated

from blood sampling. However, prevalent cases, being less susceptible to the exposure,

represents the selected healthier sub-group surviving to the beginning of the study

(selective survival) and the risk associated with mild anemia found in prevalent cases

would thereby be underestimated. In accordance with this explanation, mortality rate

over the first seven years of follow up was lower among prevalent cases of anemia than

among incident cases. A result in agreement with that of the Leiden 85 study (mortality

rates not reported): prevalent anemia (follow-up: 5 years): HR: 1.41 (1.13-1.76);

incident anemia (follow-up: ? years [< 5 years]): HR: 2.08 (1.60-2.07) [reference 11]).

Consistently, elderly subjects with thalassemia trait have a lifelong coexistence with

mild anemia and are not at risk of a shorter survival, thus tempering the difference

between mild anemia and non-anemia with respect to mortality. In fact, if the 63

participants with thalassemia trait were removed from the analysis of the H&A65-84

cohort (the 7 thalassemic subjects in the H&A 85+ are too few for any further analysis),

all HRs over 0-15, 0-7 or 8-15 years of follow-up would consistently increase with

respect to those analyzing the complete cohort.

Following the Reviewer’s request we have discussed the point raised under the

limitation part of the Discussion:

“Since prevalent cases have had the condition for some time before the study starts,

analyzing prevalent cohorts, as almost exclusively done in the available literature,

assumes that the individuals are allocated to the anemia group at the time of blood

drawing, while only prevalent cases that are alive at that time are actually included in

the analyses. These cases, being less susceptible to the exposure, represent the selected

healthier sub-group surviving to the beginning of the study (selective survival). Not

including the subset of prevalent cases at risk before baseline but that fails to survive

until the sampling date leads to a study population biased toward favorable survival and

the risk associated with mild anemia found in prevalent cases would thereby be

underestimated. In accordance with this consideration, mortality rate over the first seven

years of follow up was lower among prevalent cases of anemia than among incident

cases. A result also in agreement with that of the Leiden 85 study [11]. Consistently,

elderly participants with thalassemia trait have a lifelong coexistence with mild anemia

and are not at risk of a shorter survival, thus tempering the difference between mild

anemia and non-anemia with respect to mortality. Furthermore, the present together

with two others [11,29] are the only studies that investigated the association between

anemia and change in hemoglobin concentration with mortality also in non-anemic

cohorts”.

Under sensitivity analysis:

“If the 63 participants with thalassemia trait (62 with mild and 1 with moderate

anemia) were removed from the analysis of the H&A65-84 cohort, hazards of death

from mild anemia would consistently increase with respect to those analyzing the

complete cohort: fully-adjusted HRs (95% CIs): 1.45 (1.22-1.72) over 0-15 years, 1.78

(1.41-2.25) over 0-7 years, and 1.21 (0.94-1.57) over 8-15 years”.

- The questionnaires were administered by trained registered nurses in the H&A cohort

and by psychologists in the M80+ cohort. Would the questionnaire responses expected

to be different in the two cohorts based on the different assessors? Please also clarify

the sentence on the agreement between interviewers on medical history queries being

very high – were there more than one interviewer per individual?

Response

As extensively explained in the quoted reference 21, in the H&A65-84 study “on

average, 46 days after the blood sample collection by the nurses, a thorough home

interview was conducted by trained psychologists … . The information collected by the

psychologists was blinded to that previously gathered by the nurses and the two

interviews were used to control for the consistency of the medical histories reported by

the participants. … Agreement between comparable items of the medical histories taken

by the nurses and by the psychologists was very high (Cohen’s k between 0.84 and

0.93).” In the attempt to be concise our explanation was not clear. “Were there more

than one interviewer per individual?” Only in the H&A65-84 study: a sample (more

than seven hundred) of the initial participants in the H&A65-84 were included in the

study on the association of mild anemia with cognitive, functional, mood and QoL

outcomes (reference 21) and, after the initial interview by the nurses, they were also

interviewed by the psychologists. “Would the questionnaire responses expected to be

different in the two cohorts based on the different assessors?” No, because the

agreement between nurses and psychologists in the H&A was very high, and the

psychologists employed the same questionnaire used in the M80+ study to collect the

medical history in the H&A.

To be clearer, we have now added some of the key missing information:

“A questionnaire was administered by specifically trained registered nurses (in the

H&A) and psychologists (in H&A85+, M80+, and the H&A65-84 sample entered in the

incident study) to ascertain habits, present and past diseases, and hospital admissions.

Agreement between nurses and psychologists on medical history queries on a large

sample of participants in the H&A study was very high (Cohen’s k between 0.84 and

0.93) [21] and psychologists in the H&A employed the same questionnaire to collect the

medical history used by the psychologists in the M80+ study”.

Statistical analysis

- It is unclear whether data from the H&A 85+ cohort is analyzed with the M80+. The

previous sections do not mention the pooling of the older cohorts. It is only first

mentioned in this section that the data for the H&A 85+ cohort and the M80+ cohort

had been pooled. Please clarify specifically and earlier that pooling occurred and how,

and discuss potential impacts of pooling these two cohorts given different data

collection processes, different follow-up times etc.

Response

Under the objectives of the present study at the end of the Introduction, it is now

specified that:

“Objective of the present study … . To examine the association of mild anemia with

mortality in the old-old, the older cohorts from the H&A study (H&A85+) and from the

Monzino 80-plus study (M80+), were pooled and followed-up over a period of 11 years

(for the M80+, mortality data were available for a 15 year period)”.

At the beginning of the “Statistical analysis” we also added:

“To investigate the association of anemia/mild anemia with mortality over 11 years in

the old-old, individual participant data from the H&A85+ and M80+ cohorts were

pooled. The rationale behind this pooling was that both studies were prospective doorto-

door population-based studies in the old-old with no exclusion criteria other than age;

both had a long lasting follow-up and were conducted during more or less matching

calendar years; life expectancy was very similar; mortality data on an ongoing basis

from the Municipal Registry Offices was available for both cohorts; the modalities to

collect health-related information were very similar or identical in both”.

“discuss potential impacts of pooling these two cohorts given different data collection

processes”: actually data collection was identical in the two cohorts.

“discuss potential impacts of pooling these two cohorts given … different follow-up

times”. For both studies dates of death during the first 11 years after blood sampling

were obtained on an ongoing basis from the Municipal Registry Offices. Since for the

M80+ mortality data were available also for a further four years, we decided to truncate

the follow-up time to the shortest period (thus obtaining the same duration in both

studies) for the main analyses. We clarified this in the Methods: “To investigate the

association of anemia/mild anemia with mortality over 11 years in the old-old,

individual participant data from the H&A85+ and M80+ cohorts were pooled”.

With regard to the potential impact of pooling these two cohorts, the inspection of

results showed in supplementary S2 Table (previous S1) leaves little doubt. For the

M80+ study, we have also added a new supplementary S6 Table where the results of the

0-11 and 0-15 periods are placed side by side for inspection.

- For the analysis of the effect of mild anemia over time, how was person-time handled

in the analysis? Were individuals contributing to the 0-7 years group up until they

reached past 7 years of follow-up?

Response

Individuals who were not at risk (i.e. those who died) by the end of the first period

considered (0-7 years) were excluded from the analysis of the second period (8-15 or 8-

11 years).

We have now specified under “Statistical analysis” that

“To examine whether the effect of mild anemia was similar over time, two survival

analyses were set up: from 0 to 7 years, and, in participants who survived the first seven

years of follow up, from 8 to 15/11 years”.

- For the analysis re-defining the criteria for mild anemia, please indicate the year that

the changes in lower limits and WHO criteria were proposed.

Response

There are no reference/established criteria for “mild anemia”. Actually WHO criteria

(2011) are recent and were published after those commonly used and considered in the

present study (Dallman, 1984; Groopman and Itri, 1999; Wilson et al, [2004]). Current

debate is about the definition of the lower limit of normal hemoglobin concentration for

the diagnosis of anemia. WHO’s most commonly used definition of anemia (1968) has

been questioned by Beutler and Waalen (2006) who proposed different lower limits of

normal hemoglobin concentration for men and women according to different age groups

(20-59 and 60+ years).

We have included also this information in the manuscript under “Definitions of anemia

and mild anemia”: “Anemia was defined according to the most commonly used WHO

criteria (1968) as a hemoglobin concentration lower than 12 g/dL in women and 13

g/dL in men [22]. Along with most grading systems [23-25], mild anemia was defined

by Dallman (1984), Groopman and Itri (1999), Wilson et al. (2004) as a hemoglobin

concentration between 10.0 and 11.9 g/dL in women and 10.0 and 12.9 g/dL in men.”

Under “statistical analysis: “To investigate whether WHO criteria (1968) may have

affected the estimated effect of mild anemia on mortality, we re-evaluated this

association using slightly higher lower limits of normal hemoglobin concentration to

define anemia in white adults proposed by Beutler and Waalen in 2006 (lower than 12.2

g/dL in women and lower than 13.2 g/dL in men) [26]. We further tested this

association also using recent WHO criteria (2011) for mild anemia (lower limit of

hemoglobin concentration: 11 g/dL) [27].”

- Please indicate in the flow chart how many individuals had further blood collection.

Response

Following the Reviewer’s suggestion we have added to the flow chart the number of

participants with a second blood sampling in the H&A65-84 and M80+ cohorts.

- Were the individuals who refused blood collection at baseline different than those who

did not, aside from age and sex (e.g., comorbidities)?

Response

Probably there must be a misunderstanding. We compared participants and nonparticipants

only for age and sex simply because age and sex were/are the only two

variables available in the municipality registry offices also for non-participants (i.e.,

eligible residents who refused or were untraceable). At any rate, age and sex are two of

the main determinants of anemia and mortality and, at least for age, also of most

chronic diseases.

Having a different study aim, the eligible for the blood sub-study of the M80+ were not

all the eligible residents, rather the sub-population of those among all residents who

accepted to participate in the main study (some 90% of all the eligible residents in any

case). We also explored the distribution of dementia and cognitive function in

participants and non-participants in the blood sub-study not because these variables are

associated with mortality (a matter of fact) or anemia (under scrutiny; in the Monzino

study for example anemia was not significantly associated with an increased risk of

developing dementia [Lucca et al. Alzheimers Dement 2020; 16 (S10): abstract]), but, as

reported in the manuscript, because these were the primary research targets (in other

words, the traits under investigation), that is, those characteristics that could have

influenced the elderly subjects in the decision to take part or not in the study/sub-study

and thus possibly determine a self-selection bias. Since the entire cohort of old-old was

the result of the pooling of two cohorts (H&A85+ and M80+), the only available

information for the H&A85+ cohort and consequently for the entire 80+ cohort was that

regarding age and sex.

By the way, at initial visit in the M80+ study diabetes, hypertension, and cancer were

more prevalent among participants with blood sampling than among those without, even

though the difference was not statistically significant for diabetes (p = 0.203) and cancer

(p = 0.239).

Results

- It appears that no individuals in the M80+ cohort were excluded due to death/not

found as in the H&A cohort. If the 2,039 individuals were deemed eligible because they

were alive at first interview, please add this detail to the first box of the flow chart.

Response

The Reviewer is right: the 2,039 are the individuals alive at first interview. As explained

above (previous comment), the eligible for the blood sub-study were not all the eligible

residents, but rather the sub-population of those among all residents who accepted to

participate in the main study.

Following the Reviewer’s suggestion we have added this detail to the first box of the

flow chart.

- This section mentions that the primary research targets were dementia and cognitive

function. This is the first time the reader comes across this information. If those were

the primary aims of the original cohorts, please specify briefly under “Study settings

and participants”.

Response

Following the Reviewer’s suggestion we have specified the aim of the M80+ study

under “Study settings and participants”:

“The M80+ is a prospective, door-to-door population-based study among oldest-old

registered residents in the province of Varese, Italy (study years: 2002-2017) aimed at

investigating cognitive decline and dementia.”

- For this study, the analyses are based on the exposure of mild anemia vs non-anemia.

Table 1 should then present the baseline characteristics of individuals for each of these

exposure groups.

Response

We have changed Table 1 as requested by the Reviewer. The amount of information to

report has required splitting the Table in two: Table 1A for the H&A65-84 and Table 1B

for the [H&A85+ and M80+].

- If exposure was assessed only at baseline, it is possible that the study captured both

prevalent and incident cases of anemia. Capturing prevalent cases can be problematic

especially when assessing the association between an exposure and mortality. For

example, some individuals might have had mild anemia for several years prior to the

baseline data collection. As mentioned before, these limitations must be discussed.

Response

Please see the answer to the same previous question: “Importantly, the limitation

section should discuss how capturing prevalent cases of mild anemia (vs. incident)

might impact the results”.

- It is unclear what the time axis is for the cohort. In the text, it appears that it is

duration of follow-up in years, although the KM curves show time to death. What were

the censoring events?

Response

Sorry, the legend of the x-axis in Figure 1 was wrong. We have changed it: “Years of

observation since blood sampling (study start)”. We have also changed the Figure

legend: “Fig.1 Survival by mild anemia status in the …”.

- What was the covariate assessment window?

Response

We have added this piece of information under the “Statistical analysis”:

“… cancer, transient ischemic attack, stroke, parkinsonism, dementia. All covariates

were assessed at baseline and for participants with two blood samples also at second

sampling.

- What was the mean follow-up time?

Response

We have added the requested information under the “Results” section::

“H&A65-84 cohort

During 15 years of follow-up after blood sampling, 230 anemic (66.9%; 205 mild

anemic) and 1928 non-anemic (46.5%) individuals died (Fig 1). The median follow-up

period was 14.0 years with 50,522 person-years of observation … .

…

H&A85+ and M80+ cohorts

During 11 years of follow-up, 526 anemic (94.1%; 449 mild anemic) and 1,137 nonanemic

individuals (88.6%) died in the pooled 80+ cohort (Fig 1). The median followup

period was 3.5 years with 7,871 person-years of observation … .

…

Risk of mortality associated with mild anemia in “healthy” elderly subjects

At baseline 1093 elderly persons (mean age 74.6, 64% women) from both H&A65+ and

M80+ had no history of any of the diseases entered as confounders in multivariable

analyses. The median follow-up period was 7 years with 7,043 person-years of

observation … .

Risk of mortality associated with anemia status assessed over time (under “Study

design” or “Results”):

In H&A65-84+, a second blood sample was available for 692 subjects … . The mean

(SD) time between samplings was 2.2 (0.1) years (median follow-up: 7.0 years; personyears

of observation: 4,265; incidence mortality rate: 3.8 per 100 person-years). …

Considering the 523 subjects without anemia at first sampling in H&A65-84+ (median

follow-up: 7.0 years; person-years of observation: 3,286; incidence mortality rate: 3.3

per 100 person-years), the 27 incident cases (5.2%; mortality rate: 11.4 per 100 personyears)

… . … .

In M80+, a second blood sample was available for 366 subjects … . The mean (SD)

time between samplings was 1.7 (1.0) years (median follow-up: 3.1 years; person-years

of observation: 1,298; incidence mortality rate: 22.4 per 100 person-years). …

Considering the 279 subjects without anemia at first sampling in M80+ (median followup:

3.4 years; person-years of observation: 1,048; incidence mortality rate: 20.5 per 100

person-years), the 65 incident cases at second sampling (23.3%; mortality rate: 31.8 per

100 person-years) showed an increased risk of mortality during the following 7 years

compared to those who did not develop anemia (fully-adjusted HR: 1.47; 95% CI, 1.04-

2.07).

…

- Was the type of anemia (i.e., anemia due to specific conditions) ascertained at

baseline? If so, please indicate in the methods section. How can we be sure that the

increased risk seen for anemia type is not due to the underlying condition, and what is

the impact of adjusting for some of these comorbidities if they are also included in the

exposure definition for anemia type?

Response

Yes, anemia type was ascertained at baseline. We have added it under “Definitions of

anemia and mild anemia”:

“Anemia types were ascertained at baseline and their definitions are reported in S1

Methods”.

The underlying condition of most anemia types are not per se associated with a risk of

mortality and in a good one fourth to one third of anemias in the elderly population the

underlying causes are even still unknown (unexplained anemia). Numerous covariates

were entered in multivariable analyses because, possibly being associated with

mortality, they could explain whether the effect of mild anemia seen at univariate

analysis was independent of other possible causes of death. With regard to “the impact

of adjusting for some of these comorbidities if they [very few] are also included in the

exposure definition for anemia type” we reported among the limitations that “Extensive

adjustment may have led to underestimation of the association strength, since mild

anemia could also be an effect of underlying pathological conditions”. However,

according to Reviewer 1 this assertion “is problematic given that the authors are

interested in causal estimates. If after adjusting the estimates move towards the null,

then that is the real effect”. Based on this comment we have decided to remove this

sentence.

- With several criteria used in this study to define mild anemia, please provide the

rationale in the methods section for the choice of cut-off used to report the primary

results.

Response

As clarified above, there are no reference/established criteria for “mild anemia”: WHO

criteria (2011) are recent and were published after those commonly used and considered

in the present study (Dallman [1984], Groopman and Itri [1999], Wilson et al. [2004]).

Given that WHO (1968) is the most commonly used definition of anemia both in clinic

and research (all the 16 studies on the association of anemia and mortality used WHO

criteria for the diagnosis of anemia), actually there were only two definitions of mild

anemia: ≥ 10 g/dL of hemoglobin (Dallman [1984], Groopman and Itri [1999], Wilson

et al. [2004]) and ≥ 11 g/dL of hemoglobin (WHO [2011]). In addition to its own

merits, we chose to use the ≥ 10 g/dL cut-off level for reporting the primary results

because we had already adopted it in the previous paper on the association between

anemia and mortality over the first 3 years of follow-up in the H&A 65-84 cohort (Riva

et al. 2009, reference 16), when WHO criteria (2011) were not yet published. In any

case, by crossing the two criteria of anemia with the two of mild anemia, the reader has

the possibility to see that whatever the range of hemoglobin considered to define mild

anemia, the results found would have been more or less the same.

Under “Definitions of anemia” we have now added:

“Anemia was defined according to the most commonly used WHO criteria (1968), as …

Along with most grading system, mild anemia was defined by Dallman (1984),

Groopman and Itri (1999) and Wilson et al. (2004) as …”.

Under “Statistical analysis”:

“To investigate whether WHO criteria (1968) may have affected the estimated effect of

mild anemia on mortality, we re-evaluated this association using slightly higher lower

limits of normal hemoglobin concentration to define anemia in white adults proposed by

Beutler and Waalen in 2006 (lower than 12.2 g/dL in women and lower than 13.2 g/dL

in men) [26]. We further tested this association also using recent WHO criteria (2011)

for mild anemia (lower limit of hemoglobin concentration: 11 g/dL) [27].”

- This section mentions pooling the oldest-old cohorts because the hazard ratios were

similar. This seems like an ad-hoc decision, which should not have been based on the

HRs. Please provide a clearer rationale as to why the two cohorts were pooled in the

first place, considering that they have different follow-up times and data collection

methods.

Response

The Reviewer is perfectly right. The rationale for pooling the results of the two cohort

studies was: both were prospective population-based studies in the old-old with no

exclusion criteria other than age; both had a long lasting follow-up and were conducted

during more or less the same years; life expectancy was very similar; for both there was

access to mortality data on an ongoing basis from the Municipal Registry Offices; the

modalities to collect health-related information were very similar or identical in both.

The inspection of Table S1 actually has only confirmed post hoc that behind the pooled

results reported in Table 2 there were very similar results in both studies.

We have now modified the sentence in question under the “Results” section::

“H&A85+ and M80+ cohorts

“Risk of mortality associated with anemia and mild anemia in subjects 85 years and

older in both studies separately were quite similar (S1 Table), therefore H&A85+ and

M80+ cohorts were pooled. ”

We have added the rationale under the “Statistical analysis”:

“To investigate the association of anemia/mild anemia with mortality over 11 years in

the old-old, individual participant data from the H&A85+ and M80+ cohorts were

pooled. The rationale behind this pooling was that both studies were prospective doorto-

door population-based studies in the old-old with no exclusion criteria other than age;

both had a long lasting follow-up and were conducted during more or less matching

calendar years; life expectancy was very similar; mortality data on an ongoing basis

from the Municipal Registry Offices was available for both cohorts; the modalities to

collect health-related information were very similar or identical in both”.

- A second sample was only available for 15.4% of subjects in the H&A 65-84 cohort.

Were the individuals with a second blood sample different from those who did not have

a second blood sample?

Response

“A second sample was only available for 15.4% of subjects in the H&A 65-84 cohort”:

as we wrote under study design, “to assess change in anemia status, a stratified random

sample of individuals from H&A65-84 was recontacted during 2005-2006 [1]”.

Participants were stratified at baseline in anemic and non-anemic strata. We included all

eligible consenting anemic and a random sample of eligible non-anemic participants.

Considering that 2,306 subjects were not randomized, a second sample was available

for 31.6% of participants in the H&A 65-84. Besides, because only subjects who

accepted to be interviewed and tested (not to donate a second blood sample) entered

into the randomization, a further 1,018 subjects who refused or were not found (and

who, in any case for the most part would not be randomized because of the limited

economic resources of the study), could be removed from the denominator, as well as

161 subjects who accepted to participate but at the time of the second blood sampling

were deceased, untraceable or withdrew their consent to participate, a second sample

was available for 68.6% of participants with one blood sample.

“Were the individuals with a second blood sample different from those who did not

have a second blood sample?” Of the 4,494 subjects of the H&A 65-84 cohort, 3,802

had one blood sample and 692 two (15.4%):

H&A 65-84 1 sample 2 samples All

Participants, No. 3,802 692 4,494

Female sex, No. (%) 2,323 (61.1) 383 (55.3) 2,706 (60.2)

Age, mean (SD), years 73.6 (5.2) 73.2 (5.2) 73.5 (5.2)

Education, mean (SD), years 7.6 (3.9) 8.1 (3.8) 7.7 (3.8)

Current smokers, No. (%) 562 (14.8) 94 (13.6) 656 (14.6)

Former smokers, No. (%) 1,183 (31.2) 245 (35.5) 1,428 (31.9)

Current alcohol use, No. (%) 2,731 (72.5) 513 (74.1) 3,244 (72.7)

Former alcohol use, No. (%) 132 (3.5) 29 (4.2) 161 (3.6)

Body mass index, mean (SD) 25.0 (4.1) 24.9 (3.9) 25.0 (4.1)

Diabetes, No. (%) 358 (9.5) 63 (9.2) 421 (9.4)

Hypertension, No. (%) 2,007 (53.7) 388 (56.6) 2,395 (54.1)

Anemia and cancer cannot be matched because of the inclusion criteria of

the study on the association of mild anemia with cognitive, functional,

mood, and QoL outcomes (all participants with anemia or cancer consenting

to be interviewed and tested were included).

Key characteristics of the entire group of participants with two blood samples are

already reported in the first sentence of the “Results” section. Under “Study design” we

have now added:

“Baseline characteristics of participants with one and two samplings were for the most

comparable”.

- For the analysis capturing the second blood samples, were patient characteristics and

comorbidities measured at baseline or at the time of the second blood sample?

Response

Patients characteristics and comorbidities were assessed both at baseline and at the time

of the second blood sample. For investigating the association of incident cases and

change in anemia status (Table 4), variables entered in the multivariable analyses were

measured at the time of second follow-up.

Under “Statistical analysis” we have added:

“… cancer, transient ischemic attack, stroke, parkinsonism, dementia. All covariates

were assessed at baseline and also at second sampling for participants with two blood

samples.”

Under Table 4:

“…; F-A: "fully"-adjusted for age, sex, education, smoking, alcohol, hypertension,

diabetes, heart failure, myocardial infarction, chronic respiratory failure, chronic renal

insufficiency, TIA, stroke, cancer, dementia, hospitalization during the previous year.

All covariates were assessed at second sampling”.

- It is worrisome that anemia was measured only at baseline (a time point not defined by

any specific event), which included both prevalent and incident cases, and that the

relationship between this exposure on mortality was assessed over up to 15 years later.

Please discuss the impact of choosing an exposure definition that only captures the

baseline status without accounting for changes in the exposure status or comorbidities

over time, and includes prevalent and incident cases. Individuals included in the mild

anemia group might not have been anemic for a large portion of the follow-up time, and

vice versa.

Response

With regard to the “prevalent cases” issue, please see the answer to the same previous

question: “Importantly, the “limitations” section should discuss how capturing prevalent

cases of mild anemia (vs. incident) might impact the results”.

With regard to the failure to account for changes in the exposure status over time in the

prevalent cohort, ours is one of only two attempts [11] to investigate also whether a

change in anemia/non-anemia status would affect the risk of death in the general

population. We report below the same answer to a very similar point raised by the other

Reviewer.

In detecting and evaluating an anemia problem in a community, reference standards are

necessary, even though they may be somewhat arbitrary” (WHO 1968). Several factors

can influence the level of hemoglobin measured (for example, physiological

fluctuations of plasma volume), especially in population-based studies. Since the

diagnosis of anemia is defined as a concentration of hemoglobin of less than a

conventional lower limit of normal, physiological fluctuations of hemoglobin

concentrations can be incorrectly classified as either “anemia” (“false positive”) or

“non-anemia” (“false negative”) when hemoglobin is determined only once. Moreover,

among those diagnosed with an anemia type susceptible to treatment, a fraction can

return over time to having a normal concentration of hemoglobin when successfully

treated or the underlying causes have been eliminated. Whatever the assumption,

considering how anemia is defined and the existence of anemia types susceptible of

treatment, imply that anemia is not necessarily a chronic, everlasting condition.

Moreover, please note that the present study has investigated the association between

anemia and mortality also in prospective cohorts of young-old and old-old with two

samplings and reported the risk associated precisely with change in anemia status

(Table 4).

Following the Reviewer’s request we have now added this point to the study

limitations:

“A further potential limitation of studies assessing hemoglobin concentration on a

single occasion, is that they cannot address within-individual changes in hemoglobin

concentration over time that may potentially affect the results. However, the present

study is one of only two attempts [11] to investigate also whether a change in

anemia/non-anemia status would affect the risk of death. Generally, being anemic or

non-anemic is a rather consistent status over time at old age in the general population:

in the present study 84% of the study sample with two blood draws continued to be

anemic or non-anemic on average over two years. However, since “false positives” (due

to physiological fluctuations) and those successfully treated would decrease the

mortality rate of the anemic group in which they were classified, whereas the “false

negatives” (due to physiological fluctuations) and incident cases would increase the

mortality rate of the non-anemic group in which they were classified, the actual risk

associated with the anemia status would tend to be underestimated in subjects with a

single blood sample. Consistently, in both young-old and old-old cohorts, subjects with

two hemoglobin determinations whose anemia “resolved” (had they been successfully

treated after or “false positive” cases at first sampling) did not show a significantly

increased risk of mortality compared to those who were consistently non-anemic, a

result almost identical to that of the cited Leiden 85 study [11]) and also to that of a

retrospective study in a selected sample of patients with chronic heart failure (risk ratio

0.98 [0.73-1.36] [Tang et al. J Am Coll Cardiol 2008;51:569-576][41]). Whereas in

those who became anemic at second sampling (had they been incident cases after or

“false negatives” at first sampling) the risk was increased with respect to prevalent

cases, again in agreement with the Leiden 85 study [11]. Considering fluctuations in

concentrations over time, it should also be noted that when the upper limit of

hemoglobin concentration for anemia and/or the lower limit for mild anemia adopted

were higher (Table 2), the estimated risks for the association between anemia/mild

anemia and mortality were very similar to those of the main analysis”.

Under the Results and in Table 4 we have now added also mortality risks associated

with hemoglobin decline (per 1 g/dL decline) in the young-old and old-old without

anemia at baseline:

“In this cohort of young-old without anemia, also hemoglobin decline between

samplings was associated with a subsequent increased risk of mortality over seven years

(Table 4). … In this cohort of old-old without anemia, also hemoglobin decline between

samplings was associated with a subsequent increased risk of mortality over seven years

(Table 4)”.

Under “Discussion”:

“Consistent with two studies in a selected population of young-old [28] and 85 years old

[11], incident anemia and decline in hemoglobin concentration were significantly

associated with increased risk of mortality in both age cohorts”.

Under “Abstract-Results”:

“In participants without anemia at baseline also hemoglobin decline was significantly

associated with an increased mortality risk over seven years in both young-old and oldold”.

With regard to the failure to account for changes in comorbidities over time, we report

below the same response to a very similar point raised by the other Reviewer.

Since the comorbidities considered as covariates are all chronic diseases or have chronic

complications (e.g. stroke and myocardial infarction), the assumption is limited to the

new occurrence (incidence) of these diseases after baseline. Please also note that this

assumption as well is common to all the literature on the subject (and, incidentally, none

of the published studies has mentioned it among the limitations). However, we agree

with the Reviewer.

In the Monzino 80-plus study, beyond baseline, another 8 follow-up assessments were

available. Thus, limited to this study, it has been possible to further investigate the

influence of time-varying covariates on the association between anemia and mortality in

subjects with one (prevalent cases) as well as in subjects with two hemoglobin

determinations (incident cases and hemoglobin decline). Moreover, in subjects with two

hemoglobin determinations it was also possible to investigate how the association

between baseline anemia and mortality was affected by change in anemia status (at

second sampling) together with prevalent plus incident covariates over the following

seven years of follow-up. We have now reported the results of these analyses in a new

Table 5 (please, see the revised manuscript): all results confirmed those adjusted only

for baseline covariates previously set out in Tables 2 and 4 and in the new

Supplementary Table reporting only the results of the Monzino 80-plus study. Though

slightly, all hazard ratios are consistently higher than those reported in the above

mentioned Tables and Supplementary Table. Moreover, in the analysis where it has

been possible to account for change in both anemia status and covariates, the risk

associated with baseline anemia resulted moderately increased also with respect to the

model further adjusted for change in covariates. This finding is in agreement with the

observation that changes in anemia/non-anemia status over time would tend to

underestimate the risk of mortality associated with anemia in subjects with a single

blood sample (please see the answer to the previous comment).

Following the Reviewer’s request we have now added this point to the study

limitations:

“Another limitation, common to all the literature on the subject, is that covariates

potentially associated with death were assessed only at cohort entry, thus failing to

account for the subsequent change over time of the covariates. However, restricted to

the old-old participants in the Monzino 80-plus study, the association between anemia

and mortality could be further adjusted also for time-varying covariates and change in

anemia status over time: the results confirmed those adjusted only for baseline

covariates and showed a slightly to moderately increased risk with respect to elderly

subjects with a single blood sample and covariate assessment at entry, in agreement

with the observation that changes in anemia/non-anemia status over time would tend to

underestimate the risk assessed at baseline.”

Under “Statistical analysis” we have added:

“All covariates were assessed at baseline. Limited to the Monzino 80-plus study,

beyond baseline, another 8 follow-up assessments were available. It was thus possible

to further adjust the multivariable model also for the influence of time-varying

covariates (age, habits, and intervening comorbidities) on the association between

anemia and mortality in subjects with one (prevalent cases) as well as two hemoglobin

determinations (incident cases and hemoglobin decline). Moreover, in subjects with two

hemoglobin determinations it was also possible to investigate how the association

between baseline anemia and mortality was affected by time-varying covariates together

with anemia/non-anemia status at second sampling”.

Under “Results”, “H&A85+ and M80+ cohorts”:

“ … was found (Table 2). Limited to the M80+ cohort, mortality risk associated with

prevalent anemia and mild anemia was slightly increased when the model was further

adjusted for time-varying covariates (anemia: HR 1.59 [95% CI: 1.39-1.82] over 11

years and HR 1.64 [95% CI: 1.42-1.89] over seven years; mild anemia: HR1.57 [95%

CI: 1.36-1.80] over 11 years and HR 1.60 [95% CI: 1.38-1.85] over seven years)”.

Under “Results”, “Risk of mortality associated with anemia status assessed over time”,

at the end of the last paragraph:

“In all subjects with two hemoglobin determinations it was possible to control for both

change in the anemia/non-anemia status together with time-varying covariates: the risk

associated with baseline anemia resulted moderately increased also with respect to the

model further adjusted only for time-varying covariates (Table 5). In non-anemic

subjects with two hemoglobin determinations, the risk of mortality associated with

incident anemia or change in hemoglobin concentration over the seven years following

the second sampling was increased also when time-varying covariates were added to the

“fully”-adjusted model (Table 5).”.

Under “Discussion”:

“In the Monzino 80-plus study the risk associated with prevalent and incident anemia

was even higher when the multivariable model could be further adjusted also for timevarying

covariates and, limited to all old-old with two blood samplings, for change in

anemia status at second sampling.”.

Under “Discussion”, strength:

“No previous study has attempted to adjust also for the influence of change in

anemia/non-anemia status and time-varying covariates”.

Discussion

- The “dose-response relationships” are only stated in the abstract and discussion. This

definition of the analysis should be explicitly stated in the methods (…to assess whether

a dose-response relationship exist…”) and results. Further, it is unclear whether this

dose relationship refers to the different cut-off criteria or the comparison of HRs

between anemia and mild anemia (which, in the case of anemia, is based on very few

individuals with non-mild anemia).

Response

In order to clarify that we refer to the results reported in Figure 2 we have now added

the following sentence under the “Statistical analysis”:

“To assess whether a dose-response relationship existed, hemoglobin concentrations

were divided into categories of 1 mg/dL and p for trend analyses carried out.”

---

## [Decision Letter · Decision Letter 1]

11 Nov 2021

PONE-D-21-13069R1Mild anemia 11- to 15-year mortality risk in young-old and old-old: Results from two population-based studiesPLOS ONE

Dear Dr. Lucca,

Thank you for submitting your manuscript to PLOS ONE. After careful consideration, we feel that it has merit but does not fully meet PLOS ONE’s publication criteria as it currently stands. Therefore, we invite you to submit a revised version of the manuscript that addresses the points raised during the review process.

We look forward to receiving your revised manuscript.

Kind regards,

Laurent Azoulay, PhD

Academic Editor

PLOS ONE

Additional Editor Comments:

Thank you for your revised manuscript. There are some outstanding issues that need to be addressed, which have been detailed by the Reviewer 2. We invite you to address those issues and resubmit the manuscript.

Reviewers' comments:

Reviewer's Responses to Questions

**Comments to the Author**

1. If the authors have adequately addressed your comments raised in a previous round of review and you feel that this manuscript is now acceptable for publication, you may indicate that here to bypass the “Comments to the Author” section, enter your conflict of interest statement in the “Confidential to Editor” section, and submit your "Accept" recommendation.

Reviewer #1: All comments have been addressed

Reviewer #2: (No Response)

2. Is the manuscript technically sound, and do the data support the conclusions?

Reviewer #1: Partly

Reviewer #2: No

3. Has the statistical analysis been performed appropriately and rigorously? 

Reviewer #1: Yes

Reviewer #2: No

4. Have the authors made all data underlying the findings in their manuscript fully available?

Reviewer #1: Yes

Reviewer #2: Yes

5. Is the manuscript presented in an intelligible fashion and written in standard English?

Reviewer #1: Yes

Reviewer #2: Yes

6. Review Comments to the Author

Reviewer #1: The authors have added to the limitations section statements which highlight the potential problems. This helps the readers interpret better. I have no new comments.

Reviewer #2: Thank you for the detailed responses to the previous comments. I have a few additional comments:

1. It would be useful to clarify the time-varying analysis. The text mentions that “limited to the Monzino 80-plus study, beyond baseline, other 8 follow-up assessments were available. It was thus possible to adjust the multivariable model also for time-varying covariates […] in subjects with one (prevalent cases) as well as change in hemoglobin concentration (incident cases and hemoglobin decline).” It is unclear whether exposure was assessed at each of the 8 follow-up assessments, or “in subjects with one” (assessment? hemoglobin determination?). The text also adds “… in subjects with two hemoglobin determinations it was also possible to investigate how the association between baseline anemia and mortality was affected by time-varying covariates together with change in anemia/non-anemia status at second sampling.” Was the exposure treated as time-varying in the analysis to permit the use of time-varying covariates? Later in the text, it is mentioned that “in all subjects with two hemoglobin determinations it was possible to control for both changes in anemia/non-anemia status together with time-varying covariates […] the model further adjusted only for time-varying covariates”, which seems to suggest that the first model was adjusted for the exposure rather than the exposure being defined as time-varying?

2. In the statistical analysis section, it is mentioned that dose response was assessed and that p for trend analyses were carried out. Only one p for trend is in the Results section but none in Table 2 so it is unclear how this analysis was carried out. Further, we can see that some of the estimates and confidence intervals between each dose category are overlapping. The statement “both age groups showed a dose-response association” should thus be corrected.

3. The study reports the number of individuals who withdrew consent, could not be traced, or refused a second sample. However, it is unclear what were the censoring events during follow-up. Other than death, we would expect the follow-up time to end when an individual moved, withdrew consent, or reach the end of the study period. Please clarify in the text what were the censoring events.

4. The results were split into two periods (0-7 and 8-15). In my opinion, there is little justification for this split, especially because the mean time between samplings is reported to be 2.2 years (H&A65-84) and 1.7 years (M80+) For individuals in the second period, was their index date (start of follow-up) the start of the 8th year? We notice in the text that the median follow-up was 7.0 years for the first period and 7.4 years for the second, which suggests that the 8th year is the start of follow-up in the second period. However, the start of follow-up in the first period was at blood sample. If true, these two different definitions of index date can be problematic.

5. We notice that the median follow-up time for non-anemic patients is 4.1 years and is 7.1 years for anemic patients. However, results showed that mild anemia is associated with an increased risk of mortality. It is probable that the additional 3 years of follow-up for the anemic patients provided an “opportunity” to record more deaths in the anemic group.

6. The manuscript would benefit from a clearer organization of the methods and results. The several cohorts, time points, and analyses conducted render the manuscript difficult for the reader to follow as it currently stands.

7. PLOS authors have the option to publish the peer review history of their article (what does this mean?). If published, this will include your full peer review and any attached files.

Reviewer #1: No

Reviewer #2: No

---

## [Author Response · Author response to Decision Letter 1]

10 Dec 2021

I am not sure I have understood the difference between responding "to specific reviewer and editor comments" and the "rebuttal letter that responds to each point raised by the academic editor and reviewer(s)" that I have uploded.

---

## [Editor Report · Decision Letter 2]

14 Dec 2021

Mild anemia and 11- to 15-year mortality risk in young-old and old-old: Results from two population-based studies

PONE-D-21-13069R2

Dear Dr. Lucca,

We’re pleased to inform you that your manuscript has been judged scientifically suitable for publication and will be formally accepted for publication once it meets all outstanding technical requirements.

Kind regards,

Laurent Azoulay, PhD

Academic Editor

PLOS ONE

---

## [Editor Report · Acceptance letter]

20 Dec 2021

PONE-D-21-13069R2 

Mild anemia and 11- to 15-year mortality risk in young-old and old-old: Results from two population-based cohort studies 

Dear Dr. Lucca:

I'm pleased to inform you that your manuscript has been deemed suitable for publication in PLOS ONE. Congratulations! Your manuscript is now with our production department. 

Kind regards, 

on behalf of

Dr. Laurent Azoulay 

Academic Editor

PLOS ONE